# DiffAdapt: Difficulty-Adaptive Reasoning for Token-Efficient LLM Inference

**Xiang LIU**[H N]      **Xuming HU**[H]      **Xiaowen CHU**[H†]      **Eunsol Choi**[N†]

[H] The Hong Kong University of Science and Technology (Guangzhou)
[N] New York University
`xliu886@connect.hkust-gz.edu.cn`
† Corresponding authors

## ABSTRACT

Recent reasoning Large Language Models (LLMs) demonstrate remarkable problem-solving abilities but often generate long thinking traces whose utility is unclear. Our work aims to improve their efficiency, enabling them to reach high performance without overthinking. First, we analyze the entropy of token probabilities in reasoning traces. Across three models, we observe a consistent U-shaped entropy pattern: high entropy on easy problems despite high accuracy, low entropy on problems with medium difficulty, and high entropy on hard problems reflecting uncertainty. Specifically, we notice 22–25% entropy reduction from easy to medium difficulty regions, suggesting an overthinking phenomenon on easy instances. Building on these insights, we introduce **DiffAdapt**, a lightweight framework that selects Easy/Normal/Hard inference strategies per question based on their difficulty and reasoning trace entropy. Each inference strategy consists of a fixed prompt, temperature and maximum token length. In contrast to existing efficiency optimization methods, our approach does not fine-tune base LLM but a small probe that classifies LLM's final hidden state, allowing inexpensive adaptation. We comprehensively evaluate our method on five models and eight benchmarks. Our method achieves comparable or improved accuracy while reducing token usage by up to 22.4%, establishing a practical path toward compute-efficient reasoning.

## 1 INTRODUCTION

Large Language Models (LLMs) emerged as powerful tools for complex reasoning tasks, spanning mathematical problem-solving (Lewkowycz et al., 2022a; Pan et al., 2024), code generation (Chen et al., 2021), and logical deduction (Wei et al., 2022; TANG et al., 2026; Tang et al., 2025b). A key ingredient in this success is intermediate reasoning steps, often referred to as a "chain of thought" (CoT), before producing a final answer (Yu et al., 2024; Li et al., 2025d; Liu et al., 2024; Kong et al., 2025). However, this capability comes at a significant computational cost. Models typically generate a lengthy and elaborate chain of thought for every problem, a process sometimes called test-time scaling (Muennighoff et al., 2025; Chu et al., 2025; Tang et al., 2025a).

Always generating long traces is fundamentally inefficient. It squanders resources on simple problems, while not necessarily providing sufficient resources for truly complex tasks (Qu et al., 2025; Sui et al., 2025; Li et al., 2025b; Liu et al., 2025d;c; Zhang et al., 2023; Dong et al., 2025). In this work, we introduce a framework to bridge this gap through systematic empirical analysis and adaptive inference strategy design. We first discover a U-shaped entropy pattern: high entropy on easy problems despite high accuracy, low entropy on problems with medium difficulty, and high entropy on hard problems reflecting uncertainty. Counterintuitively, models show high uncertainty on simple problems despite achieving high accuracy.

This motivates us to design a different inference strategy per each of this three distinct difficulty regions. We develop three simple inference strategies, each equipped with different generation length (i.e., max token length), prompt and decoding hyperparameters (e.g., sampling temperature). The prompt designed for "easy" questions encourages models to answer succinctly without overthinking,

while the prompt for high difficulty questions instructs model to think carefully. We first conduct oracle experiments with these three templates. When allowed to choose an optimal strategy per question, models achieve 50% token savings while improving the model accuracy by over 10%.

Based on this observation, we introduce **DiffAdapt**, a three-stage framework that dynamically selects inference strategy rather than applying uniform reasoning budgets to all problems. Our framework operates in three stages: (1) we use a proxy model to generate training data by sampling responses and heuristically labeling them with difficulty-based strategy assignments; (2) we train a lightweight probe on the model's hidden states to predict problem difficulty; and (3) during inference, the probe dynamically selects the appropriate reasoning strategy (Easy/Normal/Hard) for each question. Compared to the training-free baseline DEER (Yang et al., 2025b), DiffAdapt achieves superior performance with up to 62% token reduction and 18% performance improvement across eight mathematical reasoning benchmarks Lightman et al. (2023); Rein et al. (2024); Wang et al. (2024); He et al. (2024); Cobbe et al. (2021) over five models DeepSeek-AI et al. (2025); Yang et al. (2025a); Liu et al. (2025a); Hou et al. (2025).

## 2 RELATED WORK

**Training-Based Budget Control.** Several methods incorporate budget control directly into the model's training phase. Huang et al. (2025); Shen et al. (2025); Liu et al. (2025b) use reinforcement learning (RL) with difficulty-aware rewards to train a model for adaptive budgeting. Similarly, Cheng et al. (2025) employ a dual-reward system based on Group-Policy Optimization (GRPO) to encourage conciseness. ThinkPrune (Hou et al., 2025) trains long-thinking LLMs via RL with token limits, using iterative pruning rounds to achieve better performance-length tradeoffs. LC-R1 (Cheng et al., 2025) addresses "invalid thinking" through GRPO-based post-training with length and compress rewards, achieving significant sequence length reduction while maintaining performance. TL;DR (Li et al., 2025c) is a training-free method that uses mixed system1 and system2 data to control the reasoning process. AdaCoT (Lou et al., 2025) framed adaptive reasoning as a Pareto optimization problem that seeks to balance model performance with the costs associated with CoT invocation. Thinkless (Fang et al., 2025) is trained under a reinforcement learning paradigm and employs two control tokens, `<short>` for concise responses and `<think>` for detailed reasoning.

**Inference-Time Budget Control.** Other methods operate purely at inference time without requiring training. Zhang et al. (2025a) define an "$\alpha$ moment" to switch from slow to fast thinking, while Zhang et al. (2025c) modify the sampling strategy to explore a continuous concept space. Yang et al. (2025b) monitor for specific transition tokens and model confidence to perform an early exit. Ma et al. (2025) propose a method to disable the reasoning process of LLMs. These training-free methods are flexible but often underperform a learned difficulty model.

**Methods Requiring Auxiliary Models.** Some approaches rely on external models to guide the LLM's reasoning process. For instance, Li et al. (2025a) train a separate BERT model to predict the remaining reasoning length and steer the generation process. Zhang et al. (2025b) employ R1-7B as a switcher model, using prompt engineering or supervised fine-tuning for strategy selection. Liang et al. (2025) utilize an MLP-based switcher with group accuracy as training labels. While these methods can achieve good performance, they introduce the overhead of running additional non-trivial models during inference, increasing computational costs and deployment complexity. Our DiffAdapt framework, in contrast, is a very small classifier integrated directly with the LLM's internal states, adding minimal latency.

## 3 CHARACTERIZING OVERTHINKING PHENOMENON

Recent work has identified that reasoning LLMs exhibit "overthinking" behavior (Ma et al., 2025; Sui et al., 2025; Qu et al., 2025), where models generate exceedingly lengthy solution when they can arrive at correct solutions much more succinctly. Building upon this observation, we analyze this phenomenon from an entropy perspective, revealing a counterintuitive pattern where models demonstrate high uncertainty.

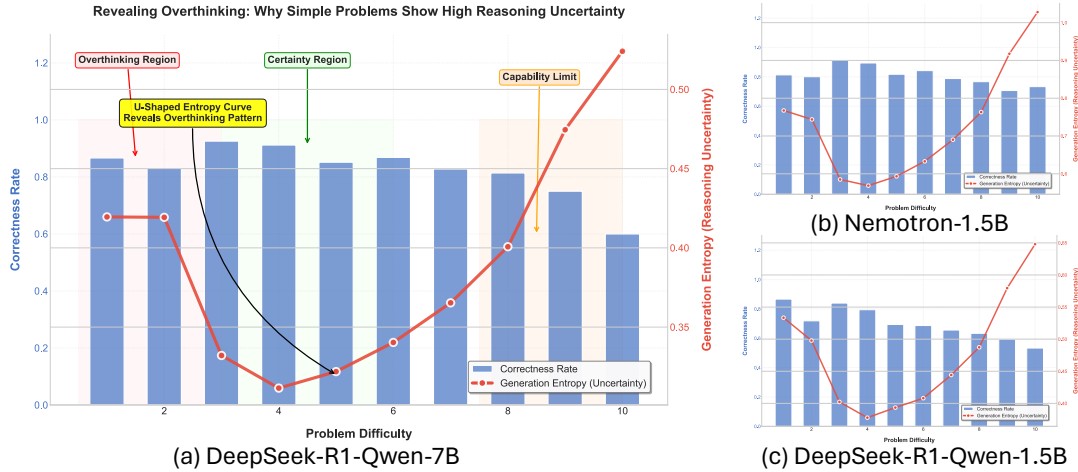

Figure 1: Visualization of model accuracy (blue bar), generation entropy (red line) per difficulty of question (x-axis), across three models (model name at the bottom of the graph). We observe a consistent U-shaped entropy curve along the difficulty levels on multiple models.

## 3.1 EXPERIMENTAL SETTING

**Dataset** We use DeepMath-103K dataset (He et al., 2025), a large-scale, challenging, decontaminated, and verifiable mathematical dataset designed for advancing reasoning capabilities. The dataset provides problems with difficulty ratings from 1-10 as evaluated by GPT-4o based on mathematical complexity and covers a wide range of mathematical topics with rigorous decontamination against numerous benchmarks. For our experiments, we created a balanced experimental set by sampling 300 questions per difficulty level, with 10 sampling iterations per problem at temperature 0.6, ensuring robust statistical analysis of entropy patterns.

**Entropy Calculation** We measure model uncertainty using generation entropy, calculated as the average entropy across all tokens in the generated sequence (Wang et al., 2025). For each token position $t$, the entropy is computed as $H_t = -\sum_{j=1}^{V} p_{t,j} \log p_{t,j}$, where $V$ is the vocabulary size and $p_{t,j}$ is the probability of predicting the $j$-th token in the vocabulary given all preceding tokens $o_{<t}$. The entropy $H_t$ corresponds to the uncertainty of the token generation distribution at position $t$, rather than an intrinsic property of the specific token $o_t$ sampled from that distribution.

**Correctness Rate Evaluation** For each difficulty rating bucket, we measure the correctness rate as the fraction of successful solutions across multiple sampling iterations. For each problem $x_i$, we generate $n$ responses $\{r_{i,j}\}_{j=1}^{n}$ and evaluate their correctness against ground truth solutions. The correctness rate for problem $x_i$ is defined as:

$$\mathcal{C}(x_i) = \frac{1}{n} \sum_{j=1}^{n} \mathbb{I}[r_{i,j} = y_i]$$

where $\mathbb{I}[\cdot]$ is the indicator function, $y_i$ denotes the ground truth solution for problem $x_i$, and $n = 10$ sampling iterations per problem in our experiments.

## 3.2 RESULTS

Figure 1 (a) illustrates this U-shaped entropy curve across different difficulty levels using the DeepSeek-R1-Distill-Qwen-7B model on the DeepMath-103K dataset (He et al., 2025). We observe consistent overthinking patterns across multiple model architectures (see Appendix F for additional models).

The U-shaped entropy curve in Figure 1 (a) reveals three distinct regions, where each region might benefit from different computational strategies:

- **Overthinking Region (Easy Problems):** High correctness rates coupled with high entropy, indicating the model is uncertain about problems it can solve well. This counterintuitive pattern

Table 1: Our three inference strategy configurations. |Max| refers to the predefined maximum token budget for generation. Full prompts and detailed configurations can be found in the Appendix C.

| Strategy | Temperature | Max Tokens | Simplified Prompt Template |
|---|---|---|---|
| **Easy** | 0.5 | $0.4 \times$ |Max| | Direct solving with verification |
| **Normal** | 0.8 | $1.0 \times$ |Max| | Step-by-step methodical approach |
| **Hard** | 0.4 | $0.5 \times$ |Max| | Resource-aware strategic reasoning |

**Performance vs. Token Consumption for Different Strategy Selections**

Table 2: Plots summarizing task performance (y-axis) and efficiency (x-axis) on Qwen3-4B model with various inference strategies. In all datasets, we find oracle strategy (i.e., the strategy that achieves the correct answer while incurring minimal tokens or the strategy incurring minimal tokens if no strategy leads to correct answer) significantly outperforms any uniform setting.

suggests unnecessary computational overhead and motivates a *Easy reasoning strategy*, which allocates minimal computational resources.

- **Certainty Region (Normal Problems):** Low entropy with optimal performance, representing the sweet spot where the model's uncertainty aligns with task complexity. These problems benefit from *Normal reasoning strategy*, which allocates standard computational resources.
- **Capability Limit Region (Hard Problems):** High entropy with low accuracy, indicating genuine difficulty where the model struggles. These problems may benefit from *Hard reasoning strategy*, which allocates less computational resources and prevents getting stuck in reasoning.

This analysis naturally leads to a three-tier adaptive strategy: **Easy**, **Normal**, and **Hard** reasoning modes, each tailored to the computational needs revealed by the overthinking phenomenon. We develop such three inference strategies and present oracle experiments with them in the next section.

## 4   ORACLE EXPERIMENTS WITH THREE INFERENCE STRATEGIES

**Inference Strategy Set** We design the inference strategies to address the challenges revealed by our overthinking analysis. For **Easy problems**, we use lower temperature and reduced tokens to prevent the model from exploring unnecessary solution paths. **Normal problems** receive standard temperature with full token budget to enable comprehensive reasoning in the optimal region where the model's uncertainty appropriately matches task complexity. For **Hard problems**, we implement a **"Fail Fast" mechanism** with strict token limits. Since our analysis indicates that these problems typically exceed the model's capabilities regardless of generation length, this strategy prioritizes cutting computational losses on likely-to-fail queries to reallocate resources, rather than engaging in potentially unproductive reasoning. Table 1 summarizes the specific configurations for each reasoning strategy. Notably, these hyperparameters were empirically determined through a systematic grid search optimization rather than selected heuristically. Further details are provided in Appendix C.

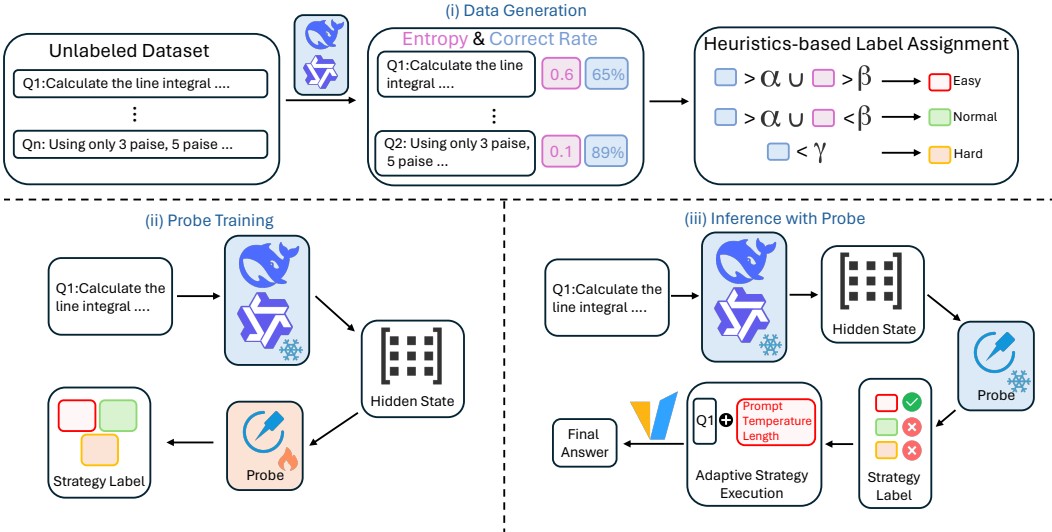

Figure 2: Overview of the DiffAdapt framework. Top: (i) **Data Generation**: We sample multiple responses from the proxy model, and compute statistics to heurstically assign inference strategy. This process yields a training dataset for the probe; (ii) **Probe Training:** We train lightweight probe which takes model's hidden states after processing the query and predict its inference strategy label; and (iii) **Inference with Probe:** where the trained probe dynamically selects appropriate inference strategies (Easy/Normal/Hard).

**Experimental Setting** To establish the theoretical upper bound of our approach, we conduct an oracle experiment using Qwen3-4B across eight reasoning benchmarks. We evaluate on five mathematical reasoning tasks: GSM8K (Cobbe et al., 2021), MATH500 (Lightman et al., 2023), AIME 2024 & 2025, and OlympiadBench (He et al., 2024), along with three out-of-domain benchmarks: Minerva (Lewkowycz et al., 2022b), GPQA (Rein et al., 2024), and MMLU-Pro (Wang et al., 2024). For each input, we generates outputs from each of the three inference strategies, **Easy**, **Normal**, and **Hard**. The Oracle strategy is determined through a two-step process: first, we identify all strategies that yield a correct answer, and from this set, we then select the one with the minimal token consumption. The parameter configurations are set as the same as the fixed strategies in Table 1, and the max token limit is 32K. More visualization details can be found in Appendix G.

**Results** Figure 2 shows that the oracle consistently dominates fixed strategies across all benchmarks, yielding a 7.2% average accuracy gain over the best fixed baseline. Token allocation adapts with problem difficulty, ranging from 198 tokens on GSM8K to 4,675 on AIME25, highlighting the need for adaptive compute budgeting. These results substantiate the overthinking hypothesis and quantify the benefits of difficulty-aware reasoning in both accuracy and efficiency. The oracle's Pareto-optimal frontier across benchmarks sets actionable performance targets for DiffAdapt.

# 5   THE DIFFADAPT FRAMEWORK

To enable LLMs to adapt their inference process based on question difficulty, we introduce a three-stage framework that leverages the model's internal representations to adapt inference to question difficulty. Figure 2 illustrates our overall approach, consisting of three sequential stages from data preparation to deployment.

## 5.1   STAGE 1: DATA GENERATION WITH PROXY MODEL SAMPLING

The first stage generates training data for the difficulty classifier through a self-supervised approach. Starting from an unlabeled dataset, we use a proxy model—typically the same LLM—to sample responses and compute its entropy and correctness.

For each problem $x$, we prompt the model to generate complete reasoning steps and final results with a maximum length of 32K tokens. We then compute the model's uncertainty using the same genera-

tion entropy calculation described in Section 3. We perform this process with 10 sampling iterations per problem to ensure robust entropy estimates. This generation entropy serves as a proxy for task complexity. Grounded in the U-shaped entropy pattern in Section 3 and the Pareto-optimal Oracle analysis (Appendix D.3), we adopt the following heuristic labeling rule: **Normal** if correctness $\geq \alpha$ and entropy $\leq \beta$; **Hard** if correctness $< \gamma$; **Easy** otherwise (typically the low-difficulty anomaly with moderate correctness but unexpectedly high uncertainty). The thresholds $\alpha, \beta, \gamma$ are set per model using a simple heuristic informed by the observed entropy–correctness distributions; we optionally perform a light sanity check on a small validation split to ensure label stability. Per-model values are reported in Appendix D.4.

## 5.2 STAGE 2: PROBE TRAINING ON HIDDEN STATES

The second stage trains a lightweight probe using the generated labeled data. As shown in the bottom-left of Figure 2, we extract hidden state representations $h_L$ from the last layer after prefilling the question.

A small probe $C$ parameterized by $\boldsymbol{\theta}$ learns to predict difficulty levels from these hidden states. The probe is implemented as a simple multi-layer perceptron (MLP):

$$d = \text{softmax}(W_2 \cdot \text{ReLU}(W_1 h_L + b_1) + b_2)$$

where $\boldsymbol{\theta} = \{W_1, W_2, b_1, b_2\}$ are the learnable parameters, and $d$ represents the predicted difficulty distribution over the three classes (Easy/Normal/Hard).

The classifier parameters are optimized by minimizing cross-entropy loss:

$$\mathcal{L}(\boldsymbol{\theta}) = -\frac{1}{N} \sum_{i=1}^{N} \log P(d_i = y_i | h_L^{(i)}; \boldsymbol{\theta})$$

We keep the base LLM weights frozen, requiring only the training of a small probe network.

## 5.3 STAGE 3: INFERENCE WITH ADAPTIVE STRATEGY EXECUTION

During the inference, the trained probe dynamically selects a reasoning strategy based on predicted difficulty, as shown in the bottom-right of Figure 2. The process consists of: **Difficulty Prediction**: Extract hidden states after prefilling the question and predict difficulty using the trained probe. **Strategy Selection**: Map predicted difficulty to corresponding reasoning strategy (Easy/Normal/Hard). **Adaptive Execution**: Apply the selected strategy with appropriate computational budget allocation.

This difficulty prediction step does not interfere with the model's prefilling and decoding processes, making it compatible with most inference optimization techniques including batching, KV cache, prefix cache, and others (Kwon et al., 2023; Zheng et al., 2024).

# 6 EXPERIMENTAL RESULTS

## 6.1 EXPERIMENTAL SETUP

**Models.** We evaluate our framework on five models: three reasoning LLMs (Qwen3-4B (Yang et al., 2025a), DeepSeek-R1-Qwen-7B (DeepSeek-AI et al., 2025), DeepSeek-R1-Llama-8B (DeepSeek-AI et al., 2025)) and three models trained with length control RL training (Nemotron-1.5B (Liu et al., 2025a),[1] ThinkPrune-7B (Hou et al., 2025)). The probe uses a simple MLP structure, trained for 100 epochs using the AdamW optimizer with learning rate 1e-3.

**Training Datasets.** The probe training dataset consists of a subset of the DeepMath-103K dataset, with 300 problems sampled per difficulty level. Data labeling is performed through Stage I of our DiffAdapt framework 5 to generate difficulty-based strategy assignment.

**Baselines.** We compare against fixed-strategy baselines that apply exclusively *Easy*, *Normal*, or *Hard* to all problems, as well as the dynamic early-exit method DEER (Yang et al., 2025b). For

---

[1]https://huggingface.co/nvidia/Nemotron-Research-Reasoning-Qwen-1.5B

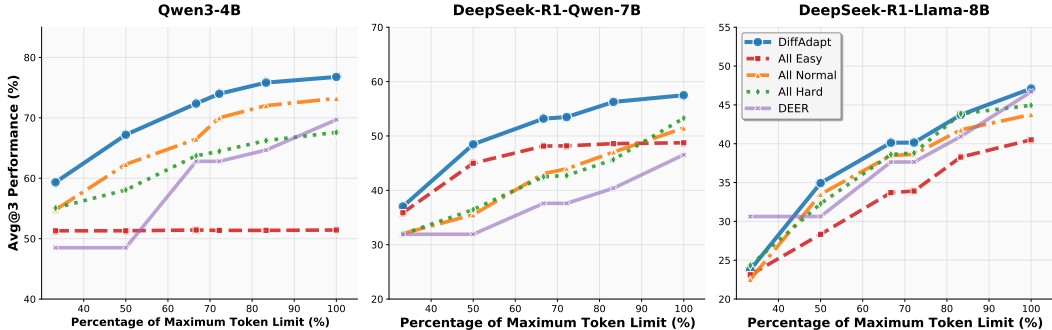

(a) In-domain performance across reasoning LLMs model architectures. DiffAdapt consistently outperforms fixed strategies across computational budgets.

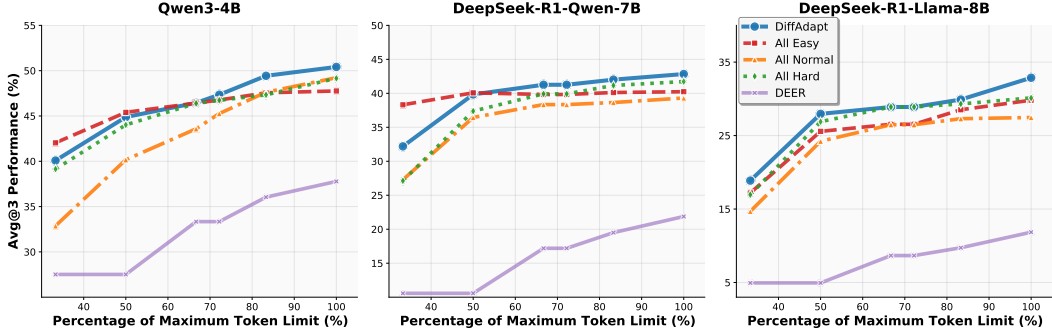

(b) Out-of-domain performance on Minerva, MMLU-Pro, and GPQA. DiffAdapt maintains effectiveness under domain shift for reasoning LLMs.

Figure 3: **Performance across reasoning LLMs model architectures and domains.** The x-axis represents different maximum token limit constraints as a percentage of the full token budget, demonstrating how different strategies perform under varying computational budgets. (a) In-domain: DiffAdapt consistently outperforms fixed strategies. (b) Out-of-domain: Effectiveness maintained under domain shift.

DEER, we align its maximum generation length with other methods and use the default think threshold of 0.9. This setup highlights the benefit of the proposed DiffAdapt framework over both static and training-free dynamic approaches.

**Evaluation Benchmarks.** We evaluate on five mathematical reasoning tasks: GSM8K (Cobbe et al., 2021), MATH500 (Lightman et al., 2023), AIME 2024 & 2025, and OlympiadBench (He et al., 2024), along with three out-of-domain benchmarks: Minerva (Lewkowycz et al., 2022b), GPQA (Rein et al., 2024), and MMLU-Pro (Wang et al., 2024).

**Experimental Design.** To demonstrate the probe's ability to perform difficulty-adaptive reasoning, we mix benchmarks of different difficulty levels. We present averaged results across benchmarks with varying domains, in-domain (GSM8K, MATH500, AIME24&25, OlympiadBench) and out-of-domain (Minerva, GPQA, MMLU-Pro) to show adaptive performance across the difficulty spectrum. Each experiment is run three times; we report the mean across runs. For evaluation, we follow reliability protocols from Ye et al. (2025)[2],Chen et al. (2025)[3]. *Max token limits*: for each model–benchmark pair, we first allow a generous cap (e.g., 32K) to observe the model's longest output and then set a nearby rounded value as the per-benchmark max_tokens used in plots; see Appendix Table 13 for the finalized values and Appendix D.6 for details.

## 6.2 REASONING LLMs RESULTS

To validate the generalizability of our DiffAdapt framework, we conducted comprehensive experiments across different reasoning LLMs model architectures and scales. Figure 3 presents the

---

[2]https://github.com/GAIR-NLP/LIMO
[3]https://github.com/IAAR-Shanghai/xVerify

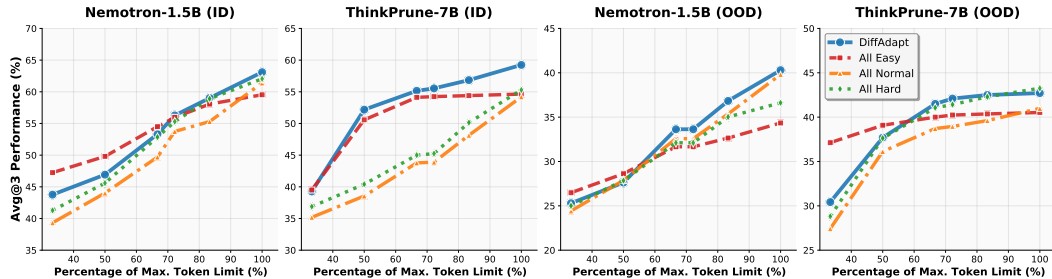

Figure 4: **DiffAdapt orthogonality with Length Control RL methods.** Performance analysis across three LC-RL trained models on both ID and OOD datasets.

performance-efficiency trade-offs for three representative models on both in-domain (ID) and out-of-domain (OOD) evaluation datasets.

**Performance Analysis** DiffAdapt consistently outperforms fixed strategies across all model architectures and domains. On in-domain tasks (Figure 3a), Qwen3-4B shows the largest improvements, particularly at higher token budgets. DeepSeek-R1-Qwen-7B demonstrates stable gains, while DeepSeek-R1-Llama-8B exhibits significant benefits across model families. On out-of-domain evaluation (Figure 3b), the framework remains effective: DiffAdapt delivers consistent improvements across models and domains, with gains becoming more pronounced as the maximum token budget increases. By contrast, the training-free dynamic baseline DEER performs comparably to the strongest fixed strategy on in-domain datasets but still lags behind DiffAdapt; under distribution shift (Minerva, MMLU-Pro, GPQA), DEER exhibits limited generalization and larger performance degradation. These results confirm DiffAdapt's robustness across different architectures, scales, and domains.

**Key Findings** The results reveal distinct performance patterns for different strategies. Easy strategies achieve high performance with minimal tokens but plateau quickly. Normal and Hard strategies improve continuously with increased computational budgets. This validates averaging across benchmarks of varying difficulty, as it captures the distinct computational requirements of different problem complexities. DiffAdapt exploits these patterns through adaptive strategy selection, consistently outperforming fixed approaches across all token limits. Compared with DEER, DiffAdapt delivers larger and more stable gains across token budgets and domains, indicating that difficulty-aware strategy selection generalizes better than confidence-based early exit. Models with **larger inter-strategy performance differences show greater DiffAdapt improvements**, confirming that adaptive selection benefits from strategy diversity.

### 6.3 ORTHOGONALITY WITH LENGTH CONTROL RL METHODS

We evaluate DiffAdapt on models trained with Length Control RL (LC-RL) to demonstrate orthogonality with existing training-based approaches. Figure 4 shows results across two LC-RL models: Nemotron-1.5B, ThinkPrune-7B on both in-domain and out-of-domain datasets.

**Key Findings.** Observing the Easy, Normal, and Hard strategy curves reveals that Easy strategies achieve superior performance in most settings, as LC-RL training adapts these models to solve problems efficiently with low computational cost. DiffAdapt shows slightly lower performance than Easy strategies under low token limits but achieves state-of-the-art results under high token budgets. This aligns with our design philosophy, enabling LLMs to maintain high performance and efficiency across problems of varying difficulty and different token budgets.

This analysis establishes that DiffAdapt can be effectively combined with existing training-based optimization methods, offering a readily integrable solution that enhances reasoning efficiency without requiring modifications to the underlying training paradigm.

### 6.4 COMPUTATIONAL EFFICIENCY ANALYSIS

We quantify computational efficiency in terms of tokens consumed and latency. For token consumption, we measure the average relative token reduction of DiffAdapt compared to a fixed Normal

Table 3: Comparing token savings (%) of our method and baseline (DEER). Negative means more tokens than the baseline.

| Model | DiffAdapt | DEER |
|---|---|---|
| DS-R1-Qwen-7B | 9.7% | −53.3% |
| Qwen3-4B | 22.4% | −27.5% |
| ThinkPrune-7B | 10.1% | – |

Table 4: End-to-end inference time comparison. All methods use vLLM backend.

| Method | Time (minutes) ↓ |
|---|---|
| vLLM Baseline | 64 |
| + DEER | 57 |
| **+ DiffAdapt** | 10 |

strategy across eight benchmarks $B$:

$$\text{Token Savings} = \frac{1}{|B|} \sum_{b \in B} \frac{\text{Tokens}_{b,\text{Normal}} - \text{Tokens}_{b,\text{Method}}}{\text{Tokens}_{b,\text{Normal}}} \times 100\%. \tag{1}$$

A higher value for Token Savings indicates greater efficiency, meaning more tokens are saved compared to the Normal strategy. Conversely, a negative value signifies that the method consumed more tokens than the baseline.

**Token cost** From Table 3, we observe substantial efficiency gains from DiffAdapt across models: Qwen3-4B achieves 22.4% token savings, while DS-R1-Qwen-7B and ThinkPrune-7B reduce usage by around 10%. In contrast, DEER *increases* token usage relative to Medium (e.g., −27.5% on Qwen3-4B and −53.3% on DS-R1-Qwen-7B). The reason is mechanistic: DEER operates under a fixed token budget and chooses continuation length primarily by confidence/probability, which frequently drives generations to the maximum token cap rather than allocating tokens adaptively by problem difficulty.

**Latency** As shown in Table 4, DiffAdapt reduces end-to-end wall-clock time by $6\times$ vs vLLM baseline and $5\times$ vs DEER (both using vLLM backend) under identical settings (Qwen3-4B; first 40 OlympiadBench problems; batch size 10; single A800 GPU; max token limit 32K; temperature 0.6; DEER think threshold 0.9), corroborating that token savings translate into practical runtime speedups.

These efficiency gains translate directly into lower inference cost at comparable accuracy levels.

# 7 ABLATION STUDIES AND ROBUSTNESS ANALYSIS

To rigorously validate the design choices and robustness of DiffAdapt, we conducted extensive ablation studies regarding hyperparameter sensitivity, probe architecture, and reasoning integrity.

## 7.1 ROBUSTNESS AND DESIGN CHOICES

We investigate three key dimensions: (1) **Threshold Sensitivity**: whether the method requires manual tuning per model; (2) **Probe Architecture**: whether a non-linear MLP is necessary; and (3) **Data Efficiency**: performance impact of reducing training data. Our evaluation protocol aligns with Section 6, utilizing in-domain benchmarks including GSM8K, MATH500, AIME 24&25, and OlympiadBench.

**Sensitivity to Thresholds** ($\alpha, \beta, \gamma$) A key concern for deployment is whether the difficulty thresholds require fine-grained calibration. To test this, we replaced the optimized thresholds of Qwen3-4B with a configuration **directly transferred from the DeepSeek-R1 model family** ($\alpha = 0.85, \beta = 0.35, \gamma = 0.60$). As shown in Table 5 ("Transferred Config"), the performance difference is negligible (avg. difference $\approx 0.3\%$). This confirms that DiffAdapt is highly robust to hyperparameter variations, enabling "plug-and-play" deployment without per-model re-calibration.

**Probe Architecture and Data Scale** Table 5 also highlights the impact of probe design. Replacing our 2-layer MLP with a simple Linear Head leads to a consistent accuracy drop ($\sim 3.2\%$), justifying the need for a lightweight non-linear classifier. Conversely, reducing the training data to only 30% results in minimal degradation, demonstrating that DiffAdapt is extremely data-efficient compared to RL-based methods that typically require large-scale rollout data.

Table 5: Ablation study on Qwen3-4B. We compare the Default DiffAdapt configuration against: (a) **Transferred Thresholds** from DeepSeek-R1 (to test robustness), (b) a **Linear Probe** (to test architecture necessity), and (c) **30% Training Data** (to test data efficiency).

| | Baseline | Robustness Check | Probe Design Ablation | |
|---|---|---|---|---|
| **Token Budget** | **DiffAdapt (Default)** | **Transferred (DeepSeek-R1)** | **Linear Head** | **30% Data** |
| 33.3% | 59.3 | 59.9 | 56.4 | 64.5 |
| 50.0% | 67.2 | 66.8 | 64.1 | 67.9 |
| 66.7% | 72.4 | 72.7 | 68.7 | 69.3 |
| 83.3% | 75.8 | 76.0 | 72.2 | 69.8 |
| 100% | 76.8 | 76.6 | 74.0 | 70.1 |
| **Average** | **70.9** | **71.2** | 67.7 | 68.5 |

Table 6: Pairwise comparison of reasoning quality (N=50). A blind Judge (Qwen3-30B) compared DiffAdapt vs. Baseline.

| Outcome | Count | Percentage |
|---|---|---|
| **DiffAdapt Wins** | **38** | **76%** |
| Baseline Wins | 6 | 12% |
| Tie | 6 | 12% |

Table 7: Failure analysis of the cases where Baseline won. "Truncation Error" indicates actual logic failure.

| Reason for Loss | Frequency |
|---|---|
| Subjective Preference | 10% (5/50) |
| **Truncation Error** | **2% (1/50)** |

## 7.2 REASONING INTEGRITY ANALYSIS

To address the concern that aggressive token reduction might compromise the logical completeness of reasoning chains (e.g., causing early truncation), we conducted a blind, pairwise **LLM-as-a-Judge** study, more details can be found in Appendix E.

We sampled 50 queries from GSM8K and used Qwen3-30B-A3B as an impartial judge to evaluate anonymized outputs from DiffAdapt and the Baseline (Normal strategy). The judge was explicitly instructed to penalize logical gaps. As shown in Table 6, DiffAdapt was preferred in **76%** of cases, with the judge often citing that DiffAdapt produced "more concise and direct" reasoning without redundancy. Notably, catastrophic failure due to early truncation occurred in only **2%** of cases (Table 7), refuting the concern that efficiency comes at the cost of reasoning integrity.

## 8 CONCLUSION

We characterize a overthinking phenomenon in LLMs, *U-shaped entropy patterns* across multiple architectures. This counterintuitive finding challenges the assumption that more computation always improves reasoning.

Our study offers three takeaways: (1) an empirical characterization of overthinking, with a consistent 22–25% entropy reduction from simple to optimal regions that reveals systematic inefficiency; (2) Oracle analysis suggesting a large potential for difficulty-aware inference strategy selection; and (3) **DiffAdapt**, a lightweight framework that predicts difficulty from hidden states and selects Easy/Normal/Hard strategies, matching or improving accuracy while reducing tokens by up to 22.4% across five models and eight benchmarks.

DiffAdapt requires no LLM retraining, is compatible with common inference optimizations (e.g., batching, KV/prefix caching), and is orthogonal to length-control RL methods. We presents a simple, lightweight solution to allow adaptive computation allocation for LLM reasoning.

## REPRODUCIBILITY STATEMENT

We aim for full reproducibility. Upon publication, we will release code, prompts, evaluation scripts, and configuration files to reproduce all tables and figures. We specify all random seeds, sampling settings (temperature/top-$p$/number of samples $n$=10), and the base token limit D.6. We will provide instructions for dataset preparation (e.g., DeepMath-103K splits and filtering), model versions used (Qwen3-4B, DeepSeek-R1-Qwen-7B, DeepSeek-R1-Llama-8B, Nemotron-1.5B, ThinkPrune-7B),

and hardware/software environments needed to replicate results. All plots in the paper are generated by released scripts.

## ETHICS STATEMENT

This work studies compute-efficient reasoning strategies for LLMs on public, decontaminated reasoning datasets. No personally identifiable or sensitive data are used. We discuss potential risks of misinterpretation and over-reliance on automatic reasoning systems and recommend careful human oversight in high-stakes scenarios. We will document evaluation limitations and known failure modes, and we avoid claims beyond the evaluated settings.

## ACKNOWLEDGEMENTS

We thank the members of NYU ML$^2$ for their helpful feedback. This work was supported in part through the NYU IT High Performance Computing resources, services, and staff expertise. The work is partially funded by NSF CAREER award 2443271 and NSF award RI-2521091; the National Natural Science Foundation of China (Grant No. 62506318 and 62272122); the Guangdong Provincial Department of Education Project (Grant No. 2024KQNCX028); the Scientific Research Projects for the Higher-educational Institutions (Grant No. 2024312096) and the Guangzhou-HKUST(GZ) Joint Funding Program (Grant No. 2025A03J3957) from the Education Bureau of Guangzhou Municipality; and Hong Kong CRF grants under Grant No. C7004-22G and C6015-23G.

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

APPENDIX

## A  USE OF LARGE LANGUAGE MODELS

We used LLMs solely to aid and polish the writing (e.g., wording refinement and grammar), without generating or altering experimental designs, methods, results, or conclusions. All technical content, analyses, figures, and tables were authored and verified by the researchers.

## B  LIMITATIONS AND FUTURE WORK

**Transferability and Calibration.** While our sensitivity analysis demonstrates that strategy thresholds are robust across model families (e.g., transferring parameters from DeepSeek-R1 to Qwen3 yields negligible $< 0.3\%$ deviation), extremely distinct domains may still benefit from a lightweight calibration phase. Future work could explore completely calibration-free mechanisms.

**Prefill-only Trade-off.** Our probe relies solely on prefill hidden states to predict difficulty. While this design minimizes inference latency by avoiding interruption of the decoding process, it inherently ignores generation-time dynamics that might reveal emerging complexity. A promising future direction is to incorporate lightweight generation signals (e.g., early step-wise entropy), though this introduces an efficiency-precision trade-off that must be carefully managed.

**Labeling via proxy models.** Our difficulty labels rely on a proxy model and sampling protocol (10 samples at temperature 0.6). Thresholds $(\alpha, \beta, \gamma)$ are tuned per model and may require re-calibration when transferring across domains or changing the sampling configuration. In deployment, we recommend a light validation phase to re-establish thresholds.

**Failure modes under tight budgets.** For particularly hard or error-prone cases, aggressive budget reduction can harm accuracy. A practical fail-safe is to fall back to the *Normal* strategy when the probe confidence is low, the prefill signal is out-of-distribution, or the selected strategy underperforms recent history.

## C  COMPLETE REASONING STRATEGY

This section provides the complete reasoning strategy configurations used in our three-tier adaptive reasoning framework. Each strategy employs specific prompts designed to guide the model's reasoning behavior according to the computational requirements identified through our overthinking analysis.

### C.1  EASY STRATEGY PROMPT

For problems identified as easy (overthinking region), we use a direct approach to minimize unnecessary computational overhead:

```
<think>

This looks straightforward. Let me solve it directly while double-checking my
approach.

</think>
```

**Configuration:**

- Temperature: 0.5 (lower randomness for direct solving)
- Max Tokens: $0.4 \times |\text{Max}|$ (reduced computational budget)
- Approach: Direct problem-solving with minimal intermediate steps

### C.2  NORMAL STRATEGY PROMPT

For problems in the optimal region, we employ standard methodical reasoning:

```
<think>
```

I'll break this down into clear, logical steps and solve methodically.

**Configuration:**

- Temperature: 0.8 (standard exploration level)
- Max Tokens: $1.0 \times |\text{Max}|$ (full computational budget)
- Top-p: 0.95 (diverse sampling for comprehensive reasoning)
- Approach: Step-by-step logical decomposition

### C.3 HARD STRATEGY PROMPT

For problems at the capability limit, we focus on efficient resource utilization and early termination of futile paths:

```
<think>
```

This appears intricate. I'll outline the main method while being mindful of computational resources.

**Configuration:**

- Temperature: 0.4 (lowest randomness for focused reasoning)
- Max Tokens: $0.5 \times |\text{Max}|$
- Approach: Strategic method outline to implement a "Fail Fast" mechanism

### C.4 DESIGN RATIONALE

The prompt design reflects our empirical findings from the overthinking analysis:

- **Easy strategy** discourages overthinking by emphasizing directness and verification rather than extensive exploration.
- **Normal strategy** encourages systematic reasoning with full computational resources for optimal performance.
- **Hard strategy** prioritizes resource conservation, identifying likely-to-fail queries early to avoid getting stuck in unproductive reasoning loops.

### C.5 HYPERPARAMETER OPTIMIZATION

To determine the optimal configuration for these strategies, we avoided heuristic selection and instead conducted a comprehensive **Grid Search** experiment on the **MATH500** dataset.

**Search Space:** We evaluated **125 distinct parameter combinations** ($5_{\text{Normal}} \times 5_{\text{Hard}} \times 5_{\text{Easy}}$), varying Temperature ($T \in [0.1, 1.2]$) and Max Token Ratios ($L \in [0.25\times, 1.0\times]$).

**Selection Criterion:** We employed a constrained optimization approach:

1. **Filter by Accuracy:** We first identified all parameter combinations that maintained high accuracy ($\geq 95\%$) on the validation set.
2. **Minimize Cost:** From these candidates, we selected the configuration that yielded the **lowest average token consumption** (960.5 tokens).

Table 8: Comparison of Top-Performing Strategy (Ours) vs. Heuristic Baselines on MATH500. Our grid-search tuned configuration achieves the best trade-off between accuracy and efficiency.

| Strategy Configuration | Description | Acc (%) | Avg Toks | Max Toks | Insight |
|---|---|---|---|---|---|
| **DiffAdapt (Ours)** | **Grid-search tuned:** Normal($T$=0.8), Hard($T$=0.4, 0.5×), Easy($T$=0.5, 0.4×) | **95.0** | **960.5** | **10,208** | **Best trade-off between creativity and stability.** |
| Baseline A (Conservative) | Heuristic uniform conservative ($T$=0.6, full length for all) | 94.0 | 1,003.2 | 12,461 | Slightly lower accuracy; higher token cost. |
| Baseline B (Aggressive) | Heuristic uniform high temp ($T$=1.2, full length for all) | 88.0 | 1,274.8 | 32,626 | Suffers from "reasoning loops" on hard queries. |
| Baseline C (Efficiency) | Heuristic aggressive pruning ($T$=0.3, 0.25× length for all) | 89.0 | 896.0 | 10,551 | Good efficiency but fails on complex reasoning tasks. |

**Result:** Table 8 demonstrates that our chosen configuration is empirically optimal, outperforming heuristic baselines in efficiency while preserving top-tier accuracy.

These strategies, combined with the corresponding sampling parameters, implement the adaptive computational allocation strategy motivated by our U-shaped entropy curve analysis.

## D ADDITIONAL EXPERIMENTAL DETAILS

### D.1 CROSS-DOMAIN GENERALIZATION

To empirically demonstrate the robustness and transferability of DiffAdapt beyond pure math reasoning, we extended our evaluation to diverse out-of-domain benchmarks, including **Minerva** (scientific reasoning), **GPQA** (graduate-level domain knowledge), and **MMLU-Pro** (comprehensive general reasoning).

**MMLU-Pro Results.** We report the detailed zero-shot transfer results on MMLU-Pro in Table 9. By using the probe and thresholds trained solely on the DeepMath dataset, DiffAdapt consistently outperforms the fixed-strategy baseline by **3-7%** across different token budgets and model architectures (DeepSeek-R1-Qwen/Llama). This confirms that the "difficulty signal" captured by our probe is generic and effectively transfers to unseen domains without re-training.

Table 9: Performance comparison on MMLU-Pro (OOD Generalization). DiffAdapt is applied zero-shot using probes trained on math data.

| Token Budget | DeepSeek-R1-Qwen-7B | | | DeepSeek-R1-Llama-8B | | |
|---|---|---|---|---|---|---|
| | DiffAdapt (%) | Baseline (%) | Improvement | DiffAdapt (%) | Baseline (%) | Improvement |
| 33.3% | **32.02** | 28.21 | +3.81% | **22.14** | 17.50 | +4.64% |
| 50.0% | **35.24** | 31.07 | +4.17% | **31.29** | 27.86 | +3.43% |
| 66.7% | **35.00** | 31.07 | +3.93% | **33.74** | 31.79 | +1.95% |
| 83.3% | **35.83** | 30.36 | +5.47% | **35.45** | 32.14 | +3.31% |
| 100% | **36.48** | 30.71 | +5.77% | **39.90** | 32.50 | +7.40% |

### D.2 COMPARISON WITH "WHEN-TO-THINK" BASELINES

We further compared DiffAdapt against specialized "when-to-think" methods like ThinkLess. Unlike these methods which typically require expensive two-stage training (SFT + RL), DiffAdapt is a training-free, plug-and-play approach for the LLM.

**Setup.** We applied DiffAdapt to the **ThinkLess Stage-1 model** ('TL-1.5B-Warmup') and compared it against their fully trained **Stage-2 RL model** ('TL-1.5B-RL').

**Results.** Table 10 presents the results on MATH500 and GSM8K.

- **Efficiency:** On GSM8K, DiffAdapt consistently outperforms the RL baseline while using fewer tokens.

- **Cost-Effectiveness:** On MATH500, while the RL model achieves higher peak accuracy, DiffAdapt outperforms the Warmup baseline by significant margins (+4-8%) and achieves

competitive performance to the RL model in low-resource regimes using ~**35-50% fewer tokens**, without requiring any RL training.

Table 10: Comparison against ThinkLess (TL) baselines. DiffAdapt is applied to the TL-1.5B-Warmup model.

| | MATH500 | | | GSM8K | | |
|---|---|---|---|---|---|---|
| Budget | TL-Warmup + DiffAdapt | TL-1.5B-RL | Analysis | TL-Warmup + DiffAdapt | TL-1.5B-RL | Analysis |
| 33.3% | **64.4%** (851 tok) | 55.6% (1039 tok) | **+8.8%** / Less Toks | **67.0%** (336 tok) | 66.1% (409 tok) | **+0.9%** |
| 50.0% | **67.6%** (954 tok) | 63.3% (1460 tok) | **+4.3%** / -35% Toks | **77.0%** (357 tok) | 72.0% (465 tok) | **+5.0%** |
| 72.2% | 68.0% (973 tok) | **69.2%** (1775 tok) | Comparable / -50% Toks | **78.2%** (361 tok) | 74.3% (510 tok) | **+3.9%** |
| 100% | 68.3% (1003 tok) | **73.5%** (2020 tok) | RL peaks higher | 78.6% (364 tok) | **78.8%** (573 tok) | Comparable |

## D.3 ORACLE EXPERIMENT DETAILED RESULTS

This subsection provides the complete numerical results from our Oracle experiment across eight reasoning benchmarks. Table 11 shows the accuracy and average token consumption for each strategy on every benchmark.

Table 11: Detailed Oracle Experiment Results Across Reasoning Benchmarks With Qwen3-4B

| Benchmark | Easy Strategy | | Normal Strategy | | Hard Strategy | | Oracle Selection | |
|---|---|---|---|---|---|---|---|---|
| | Acc (%) | Tokens | Acc (%) | Tokens | Acc (%) | Tokens | Acc (%) | Tokens |
| GSM8K | 89.90 | 169.93 | 93.20 | 561.80 | 93.50 | 511.55 | 96.20 | 197.96 |
| MATH | 82.80 | 718.33 | 96.20 | 2662.06 | 93.40 | 2424.69 | 98.00 | 1278.90 |
| GPQA | 46.46 | 812.55 | 50.51 | 3022.33 | 47.98 | 2952.44 | 70.20 | 1001.14 |
| MMLU-Pro | 60.36 | 526.14 | 65.36 | 2076.16 | 62.50 | 1873.14 | 74.64 | 690.04 |
| Minerva | 46.69 | 468.74 | 55.15 | 2795.14 | 53.31 | 2248.22 | 65.81 | 866.31 |
| OlympiadBench | 50.96 | 1628.91 | 73.63 | 6722.40 | 68.59 | 5895.97 | 76.89 | 2756.25 |
| AIME 2024 | 16.67 | 3504.90 | 60.00 | 10672.23 | 46.67 | 10166.87 | 66.67 | 4429.37 |
| AIME 2025 | 16.67 | 2442.83 | 53.33 | 13733.47 | 40.00 | 10634.77 | 56.67 | 4675.00 |

**Key Observations.** The detailed results reveal several important patterns: (1) **Strategy Distribution**: Across all problems, 82.3% benefit from Easy strategy, 7.7% from Normal strategy, and 10.0% from Hard strategy, confirming the prevalence of overthinking in current reasoning approaches. (2) **Benchmark Characteristics**: Mathematical competition problems (AIME 2024/2025) require the highest computational resources, while basic arithmetic (GSM8K) achieves optimal performance with minimal tokens. (3) **Universal Improvement**: Oracle selection achieves higher accuracy than any fixed strategy across all benchmarks while maintaining efficient token usage. (4) **Efficiency Gains**: The Oracle demonstrates substantial token savings compared to always using Normal or Hard strategies, with efficiency improvements ranging from $3\times$ (GSM8K) to $5\times$ (AIME series). These results provide the empirical foundation for our DiffAdapt framework and establish clear performance targets for practical adaptive reasoning systems.

## D.4 MODEL-SPECIFIC THRESHOLD VALUES

This subsection provides the specific threshold values used for difficulty classification across different models in our framework. The thresholds $\alpha$ (correctness), $\beta$ (entropy), and $\gamma$ (correctness) are determined empirically for each model to optimize the strategy assignment performance.

Table 12: Model-Specific Threshold Values for Difficulty Classification

| Model | $\alpha$ (Normal) | $\beta$ (Entropy) | $\gamma$ (Hard) |
|---|---|---|---|
| DeepSeek-R1-Qwen-7B | 0.85 | 0.35 | 0.60 |
| DeepSeek-R1-Llama-8B | 0.85 | 0.35 | 0.60 |
| Qwen3-4B | 0.88 | 0.32 | 0.65 |

**Threshold Selection.** We use a heuristic procedure guided by the entropy–correctness scatter of each model (no exhaustive search). Concretely, we pick $\alpha$ near the knee where high-correctness,

low-entropy points concentrate; choose $\beta$ at the elbow separating low vs. high-uncertainty regimes; and set $\gamma$ to capture the reliability drop-off region in correctness. We optionally verify stability with a small validation split. This selection ensures that:

- **Normal threshold** ($\alpha$): Captures problems where the model performs consistently well with low uncertainty

- **Entropy threshold** ($\beta$): Distinguishes between confident and uncertain predictions

- **Hard threshold** ($\gamma$): Identifies problems beyond the model's reliable capability range

These model-specific thresholds reflect the inherent differences in reasoning capabilities and uncertainty patterns across different architectures and scales.

### D.5 DETAILED ALGORITHMIC PROCEDURES

This subsection provides the complete algorithmic descriptions for the three main stages of our DiffAdapt framework. These algorithms detail the implementation procedures that correspond to the conceptual framework presented in Section 5.

---

**Algorithm 1** Data Generation with Proxy Model Sampling

---

1: **Input:** Set of problems $\mathcal{X} = \{x_i\}_{i=1}^N$, LLM, thresholds $\alpha$, $\beta$, $\gamma$
2: Initialize labeled dataset $\mathcal{D} \leftarrow \emptyset$
3: **for all** problem $x$ in $\mathcal{X}$ **do**
4:      Generate $n = 10$ complete reasoning sequences with max length 32K
5:      Compute correctness rate $\mathcal{C}(x)$ and average entropy $\bar{H}(x)$
6:      **if** $\mathcal{C}(x) > \alpha$ **and** $\bar{H}(x) < \beta$ **then**
7:          $y_{\text{label}} \leftarrow$ 'Normal'
8:      **else if** $\mathcal{C}(x) < \gamma$ **then**
9:          $y_{\text{label}} \leftarrow$ 'Hard'
10:      **else**
11:          $y_{\text{label}} \leftarrow$ 'Easy'                        ▷ Overthinking cases
12:      **end if**
13:      Add $(x, y_{\text{label}})$ to $\mathcal{D}$
14: **end for**
15: **Output:** Labeled dataset $\mathcal{D}$

---

### D.6 MAXIMUM TOKEN LIMITS PER MODEL AND BENCHMARK

This subsection reports the maximum token limits used for each model on each benchmark and describes how they were determined. For each model–benchmark pair, we first ran the model under a generous cap (e.g., 32K tokens) to observe its longest response length in a less constrained setting. We then selected a nearby rounded integer as the per-benchmark max_tokens used in our analyses. This procedure standardizes evaluation across tasks and enables percentage-based truncation in Figures 3 and 4.

**Procedure example.** With a 32K cap, Qwen3-4B produced longest responses of approximately 1,500 tokens on GSM8K and 18,000 tokens on AIME24; we therefore set max_tokens to 1,500 and 18,000 for those benchmarks, respectively. Analogous rounding was applied to all model–benchmark pairs (see Table 13).

## E REASONING INTEGRITY ANALYSIS

A primary concern with efficiency-oriented reasoning methods is the potential risk of compromising reasoning integrity—specifically, whether aggressive token reduction leads to early truncation or logical gaps. To rigorously evaluate this, we conducted a blind, pairwise **LLM-as-a-Judge** study.

Table 13: Maximum token limits (in tokens) per model and benchmark (ID and OOD). Values are rounded from observed maxima under a large cap (e.g., 32K).

| Model | GSM8K | MATH | AIME 2024 | AIME 2025 | OlympiadBench | Minerva | MMLU-Pro | GPQA |
|---|---|---|---|---|---|---|---|---|
| Qwen3-4B | 1500 | 12000 | 18000 | 18000 | 15000 | 3500 | 3000 | 4000 |
| DeepSeek-R1-Qwen-7B | 500 | 3000 | 15000 | 16000 | 5500 | 1750 | 3000 | 5500 |
| DeepSeek-R1-Llama-8B | 700 | 3000 | 14000 | 14000 | 5500 | 1750 | 1750 | 3000 |
| Nemotron-1.5B | 3500 | 4000 | 7000 | 6000 | 5500 | 5000 | 3500 | 5000 |
| ThinkPrune-7B | 500 | 3000 | 15000 | 14000 | 5500 | 1750 | 2500 | 4500 |

## E.1 EXPERIMENTAL SETUP

We randomly sampled $N = 50$ queries from the GSM8K test set. For each query, we generated two responses:

- **System A (DiffAdapt)**: Our proposed method with adaptive strategy selection.

- **System B (Baseline)**: The standard Normal strategy (Temperature=0.8, full token budget).

We employed **Qwen3-30B-A3B** as an impartial judge. To ensure fairness, the evaluation was **blind** (model identities were anonymized) and **pairwise** (side-by-side comparison). The judge was explicitly instructed to evaluate based on logical completeness, coherence, and conciseness, and to penalize any instances of unjustified truncation.

## E.2 EVALUATION PROMPT

The specific prompt used for the LLM-as-a-Judge evaluation is provided below. It explicitly asks the judge to focus on the preservation of coherent reasoning under token constraints.

```
LLM-as-a-Judge Prompt

You will compare two systems on the same GSM8K math word problem.

Problem:
{problem}

Ground-truth solution (for verification only):
{ground-truth solution}

System A:
- Strategy: {strategy_A}
- Tokens used: {tokens_A}
- Final prediction: {Correct/Incorrect}
Reasoning trace:
<<<
{reasoning_trace_A}
>>>

System B:
- Strategy: {strategy_B}
- Tokens used: {tokens_B}
- Final prediction: {Correct/Incorrect}
Reasoning trace:
<<<
{reasoning_trace_B}
>>>

Decide which reasoning trace better preserves coherent, logically
complete reasoning under tight token budgets. Explain your
reasoning and output the winner (System A, System B, or Tie).
```

## E.3 RESULTS AND ANALYSIS

Table 14 summarizes the results of the blind evaluation.

Table 14: Blind pairwise comparison of reasoning quality (N=50) by Qwen3-30B Judge.

| Outcome | Count | Percentage | Judge's Common Rationale |
|---|---|---|---|
| **DiffAdapt Wins** | **38** | **76%** | "More direct," "Avoids unnecessary repetition," "Efficient logic" |
| Baseline Wins | 6 | 12% | "More detailed explanation" (in 5 cases), "Truncation" (in 1 case) |
| Tie | 6 | 12% | "Both reasoning paths are identical" |

**Frequency of Logical Failure** We performed a manual failure analysis on the 6 cases where the Baseline won:

- **Subjective Preference (5 cases)**: The Baseline produced a more verbose explanation which the judge preferred, even though DiffAdapt's response was correct and logically complete.
- **Truncation Error (1 case)**: Only a single instance (2%) involved actual logical failure due to aggressive token reduction (misclassified as Easy).

This low failure rate (2%) confirms that DiffAdapt's fallback mechanisms (Normal strategy for ambiguous cases) effectively preserve reasoning integrity while significantly reducing computational cost.

## F ADDITIONAL OVERTHINKING ANALYSIS ACROSS MODEL ARCHITECTURES

To demonstrate the universality of the overthinking phenomenon, we present additional overthinking analysis results for two more model architectures: DeepSeek-R1-Distill-Qwen-1.5B and Nemotron-Research-Reasoning-Qwen-1.5B. These results complement the main analysis presented in Section 3 and provide further evidence that the U-shaped entropy pattern is consistent across different model sizes and architectures.

### F.1 DEEPSEEK-R1-DISTILL-QWEN-1.5B OVERTHINKING ANALYSIS

Figure 5 shows the overthinking analysis for the DeepSeek-R1-Distill-Qwen-1.5B model. Despite being a smaller 1.5B parameter model, it exhibits the same characteristic U-shaped entropy curve:

- **Simple Problems (Difficulty 1-2)**: High entropy with good correctness, indicating overthinking behavior
- **Certainty Region (Difficulty 3-6)**: Reduced entropy with maintained performance
- **Difficult Problems (Difficulty 8+)**: Increased entropy with declining performance

The entropy reduction of 23.3% from simple to optimal regions demonstrates strong overthinking evidence, consistent with our findings across model architectures.

### F.2 NEMOTRON-RESEARCH-REASONING-QWEN-1.5B OVERTHINKING ANALYSIS

Figure 6 presents the analysis for Nemotron-Research-Reasoning-Qwen-1.5B, another 1.5B parameter reasoning model. This model shows the most pronounced U-shaped pattern:

- **Simple Problems (Difficulty 1-2)**: High entropy with strong correctness, showing clear overthinking
- **Certainty Region (Difficulty 3-6)**: Significantly reduced entropy with peak performance
- **Difficult Problems (Difficulty 8+)**: Highest entropy with declining accuracy

This model demonstrates a 21.3% entropy reduction from simple to optimal regions, providing additional validation of the overthinking phenomenon across different reasoning architectures.

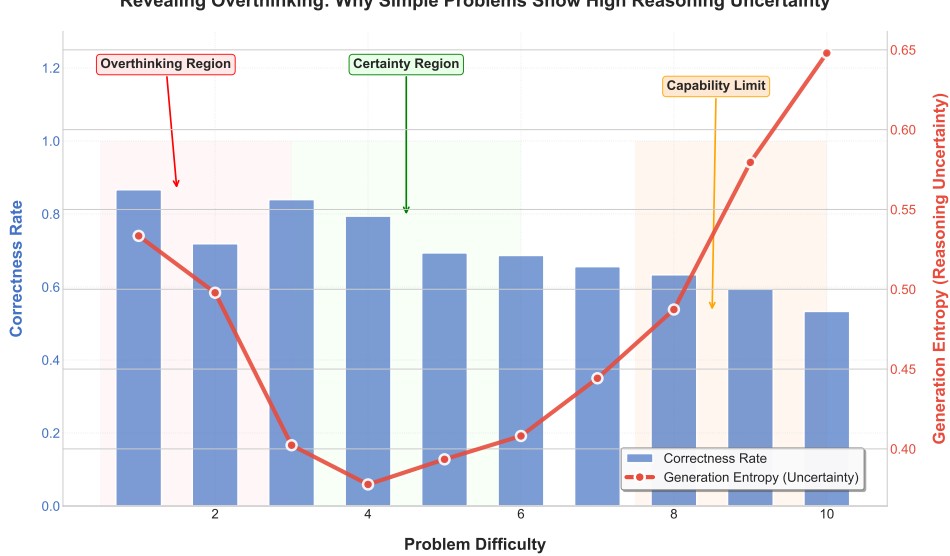

Figure 5: Overthinking phenomenon in DeepSeek-R1-Distill-Qwen-1.5B model showing the characteristic U-shaped entropy pattern across difficulty levels.

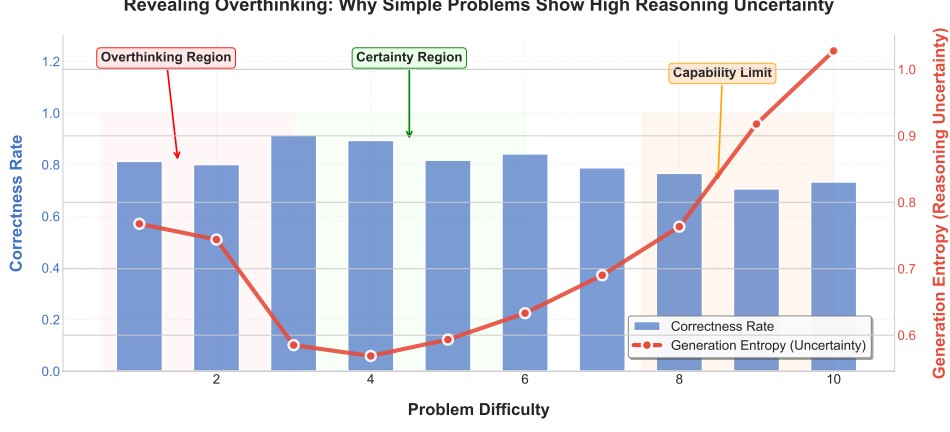

Figure 6: Overthinking phenomenon in Nemotron-Research-Reasoning-Qwen-1.5B model demonstrating the universal U-shaped entropy pattern.

# G  ADDITIONAL ORACLE ANALYSIS RESULTS

This section presents comprehensive Oracle analysis results across multiple model architectures to validate the generalizability of our findings. We conduct the same Oracle experiment described in Section 3 on three additional models: DS-Qwen-7B, Nemotron-1.5B, and DeepSeek-R1-Llama-8B. These models represent different scales, architectures, and training methodologies, providing robust evidence for the universal applicability of adaptive reasoning strategies.

## G.1 DeepSeek-R1-Qwen-7B Oracle Analysis

Figure 7 shows the performance-token trade-offs for DeepSeek-R1-Qwen-7B across all eight reasoning benchmarks. The results demonstrate consistent Oracle superiority with an average improvement of +12.3% over the best fixed strategy, validating our findings across larger model scales.

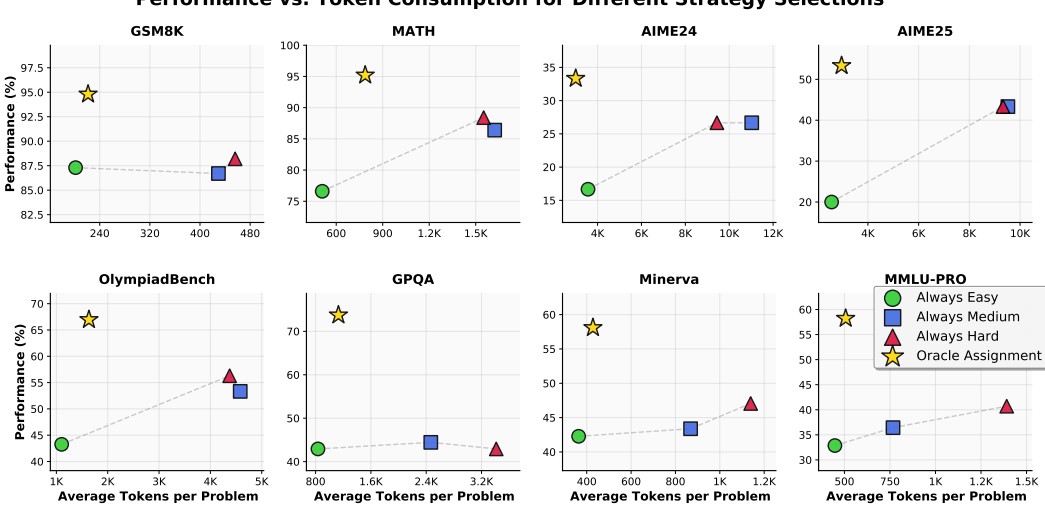

Figure 7: DeepSeek-R1-Qwen-7B Oracle Analysis: Performance vs. Token Consumption Trade-offs. The Oracle strategy (gold stars) consistently outperforms all fixed strategies across mathematical reasoning tasks (GSM8K, MATH), competition problems (AIME24/25, OlympiadBench), and out-of-domain benchmarks (GPQA, Minerva, MMLU-Pro), achieving optimal Pareto efficiency with an average +12.3% accuracy improvement.

## G.2 Nemotron-1.5B Oracle Analysis

Figure 8 presents the Oracle analysis for Nemotron-1.5B, demonstrating that adaptive strategy selection benefits extend to smaller model scales. Despite the reduced parameter count, the Oracle achieves +7.9% average improvement while maintaining superior token efficiency.

## G.3 DeepSeek-R1-Llama-8B Oracle Analysis

Figure 9 shows the most compelling results from DeepSeek-R1-Llama-8B, which achieves the highest Oracle benefits with +16.2% average improvement. This model demonstrates exceptional token efficiency, with Oracle strategy consuming significantly fewer tokens while achieving superior performance across all benchmarks.

## G.4 Cross-Model Oracle Analysis Summary

Table 15 summarizes the Oracle analysis results across all four models, demonstrating the universal effectiveness of adaptive strategy selection.

Table 15: Cross-Model Oracle Analysis Summary

| Model | Parameters | Avg. Accuracy Improvement | Dominance Rate | Token Efficiency |
| --- | --- | --- | --- | --- |
| Qwen3-4B | 4B | +7.2% | 100% | Mixed |
| DS-Qwen-7B | 7B | +12.3% | 100% | Moderate |
| Nemotron-1.5B | 1.5B | +7.9% | 100% | High |
| DeepSeek-R1-Llama-8B | 8B | +16.2% | 100% | Excellent |

**Key Insights:**

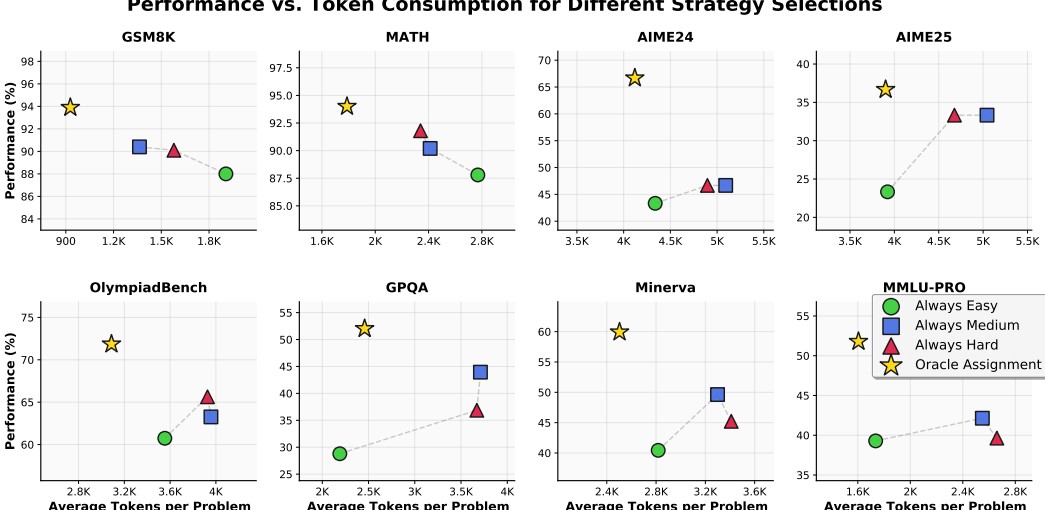

Figure 8: Nemotron-1.5B Oracle Analysis: Performance vs. Token Consumption Trade-offs. Even at smaller scale (1.5B parameters), the Oracle strategy demonstrates consistent advantages across all benchmarks, achieving +7.9% average accuracy improvement with efficient token utilization, confirming the scalability of adaptive reasoning approaches.

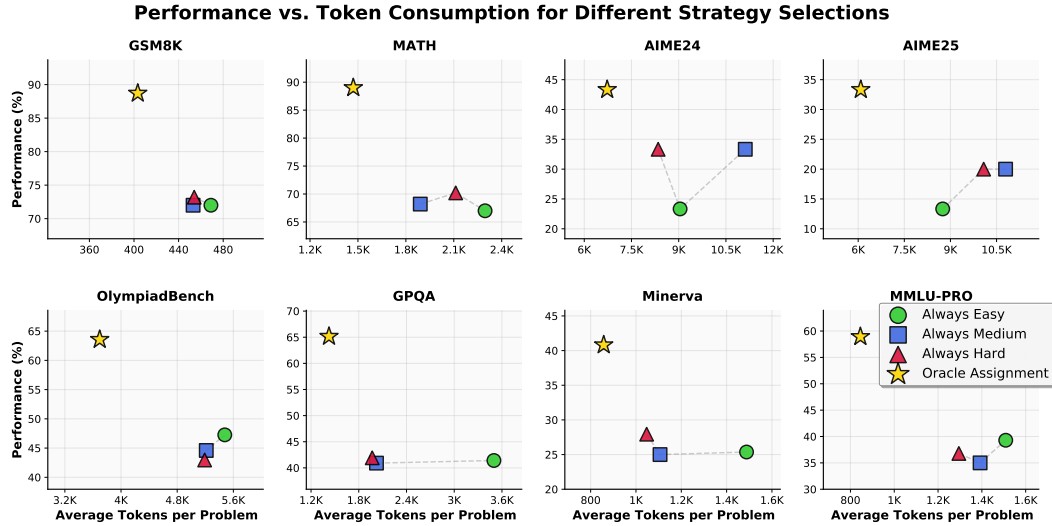

Figure 9: DeepSeek-R1-Llama-8B Oracle Analysis: Performance vs. Token Consumption Trade-offs. This model shows the strongest Oracle benefits with +16.2% average accuracy improvement and exceptional token efficiency. The Oracle strategy achieves superior performance while consuming 10-35% fewer tokens than fixed strategies, demonstrating optimal resource utilization.

- **Universal Dominance**: Oracle strategy achieves 100% dominance rate across all models and benchmarks

- **Scalable Benefits**: Performance improvements scale with model capability, ranging from +7.2% to +16.2%

- **Consistent Token Efficiency**: All models show improved resource utilization with adaptive strategy selection

- **Robust Generalization**: Benefits span mathematical reasoning, competition problems, and out-of-domain tasks

These comprehensive results provide strong empirical evidence that adaptive reasoning strategies offer universal benefits across diverse model architectures, scales, and problem domains, directly motivating the design and deployment of our DiffAdapt framework.

