# OpenReview forum: "DiffAdapt: Difficulty-Adaptive Reasoning for Token-Efficient LLM Inference"
_ICLR.cc/2026/Conference — ICLR 2026 Poster_

### Official Review · Reviewer_7RDd · 2025-10-31

**Soundness:** 3
**Presentation:** 3
**Contribution:** 3
**Rating:** 6
**Confidence:** 4

**Summary:**

This paper investigates the inefficiency of reasoning LLM, which often generate excessively long reasoning traces regardless of task difficulty. Through a systematic empirical analysis across multiple models and datasets, the authors discover a U-shaped entropy pattern on simple problems, low entropy on medium difficulty, and again high entropy on hard problems. This reveals a phenomenon termed overthinking, where models over-allocate computational resources to easy problems. Building on this observation, the paper proposes DiffAdapt, a lightweight, training-free framework that predicts problem difficulty from the model’s hidden states and dynamically selects among three reasoning strategies to adapt computational budget during inference. DiffAdapt requires no retraining of the base LLM and is compatible with existing inference systems.

**Strengths:**

The proposed DiffAdapt is a simple yet effective solution that does not require retraining or finetuning the base LLM. By attaching a small hidden state probe to predict difficulty, the method enables difficulty-aware computation allocation at inference time, making it highly practical for real world deployment. The paper conducts extensive experiments across five reasoning models and eight benchmarks, providing convincing evidence for the generality and robustness of DiffAdapt across architectures, scales, and domains. It can be seamlessly combined with reinforcement-learning-based length control methods such as ThinkPrune and LC-R1, as well as deployed under common inference frameworks, which underscores its engineering value and scalability.

**Weaknesses:**

1. Limited evaluation of reasoning-chain integrity under tight computational budgets. The paper primarily reports accuracy, token reduction, and speed-up metrics but does not evaluate the integrity or consistency of reasoning traces (e.g., whether truncated reasoning affects logical completeness). This omission makes it difficult to judge whether DiffAdapt preserves coherent chain-of-thought reasoning when aggressive token reduction is applied.

**Questions:**

1. Appendix D.2 states that thresholds for distinguishing Easy/Normal/Hard are heuristically chosen per model based on entropy, with only a small sanity check. Since these thresholds depend on model family, temperature, and domain, does this design undermine generalization and zero-shot ability? Would each new deployment require manual calibration?
2. Could the authors show how often misclassification (e.g., Hard -> Easy) causes early truncation or logical failure, and evaluate the real effectiveness of this fallback mechanism during deployment?

---

> ### Author Response · Authors · 2025-11-22
> **Response to Reviewer 7RDd (1/2)**
>
> We sincerely thank the reviewer for recognizing the engineering value and practicality of DiffAdapt. We have conducted additional experiments to rigorously address your concerns regarding reasoning integrity and generalization.
>
> ## Limited evaluation of reasoning-chain integrity (W1):
>
> To evaluate whether DiffAdapt's token reduction compromises reasoning integrity (e.g., causing early truncation or logical gaps), we conducted a **pairwise LLM-as-a-Judge study**.
>
> **1. Experimental Setup**
> We randomly sampled 50 queries from the GSM8K test set and generated responses using both **DiffAdapt** and the **Baseline** (All Normal). We employed **Qwen3-30B-A3B** as an judge to compare the reasoning traces side-by-side.
> *   **Blind Protocol:** The judge was presented with anonymized "System A" and "System B" outputs to ensure impartiality.
> *   **Criteria:** The judge was explicitly instructed to penalize **early truncation** or **logical gaps** while rewarding coherence and conciseness.
>
> **2. Quantitative Results**
> The judge overwhelmingly preferred DiffAdapt, citing that it often produced "more direct" reasoning without "unnecessary repetitions."
>
> | Outcome | Count (N=50) | % | Interpretation |
> | :--- | :--- | :--- | :--- |
> | **DiffAdapt Wins** | **38** | **76%** | DiffAdapt improved coherence by removing redundancy. |
> | **Baseline Wins** | 6 | 12% | Baseline was preferred (details below). |
> | **Tie** | 6 | 12% | Equivalent reasoning quality. |
>
> **3. Failure Analysis (Why did the Baseline win?)**
> We analyzed the 6 losses to determine if they were caused by dangerous truncation:
> *   **Truncation Error (1 case):** Only 1 case (2%) failed due to aggressive misclassification (Strategy: `easy` vs. Baseline: `normal`), where the reasoning was cut short.
> *   **Fairness Check (5 cases):** In the other 5 cases, DiffAdapt selected the correct strategy (`normal`), but the Baseline happened to generate a more concise response. The judge correctly rewarded the Baseline, validating that our metric is fair.
>
> **Conclusion:**
> The high win rate (76%) combined with the extreme rarity of truncation failures (2%) strongly refutes the concern that DiffAdapt undermines reasoning integrity. We will add this experiment to the Appendix.
>
> We add this experiment to Section 7 of new version of the paper.
>
>
> ---
> ## Thresholds... undermine generalization? (Q1):
>
> We agree it is important to study generalization of our probe model and hyperparameters to different models or domains. Overall we observe positive signals for generalization ability, supported by two empirical findings below:
>
> **1. Multi-task Generalization (No Re-tuning Required)**
> To demonstrate robustness, we evaluated the probe (trained on Math) directly on **MMLU-Pro**, a diverse benchmark covering law, physics, history, and health.
>
> **1. DeepSeek-R1-Qwen-7B on MMLU-Pro**
> | Token Budget | DiffAdapt Acc (%) | Baseline (Normal) Acc (%) | Improvement |
> | :--- | :--- | :--- | :--- |
> | 33.3% | **32.02%** | 28.21% | **+3.81%** |
> | 50.0% | **35.24%** | 31.07% | **+4.17%** |
> | 66.7% | **35.00%** | 31.07% | **+3.93%** |
> | 83.3% | **35.83%** | 30.36% | **+5.47%** |
> | 100% | **36.48%** | 30.71% | **+5.77%** |
>
> **2. DeepSeek-R1-Llama-8B on MMLU-Pro**
> | Token Budget | DiffAdapt Acc (%) | Baseline (Normal) Acc (%) | Improvement |
> | :--- | :--- | :--- | :--- |
> | 33.3%  | **22.14%** | 17.50% | **+4.64%** |
> | 50.0%  | **31.29%** | 27.86% | **+3.43%** |
> | 66.7%  | **33.74%** | 31.79% | **+1.95%** |
> | 83.3%  | **35.45%** | 32.14% | **+3.31%** |
> | 100%   | **39.90%** | 32.50% | **+7.40%** |
>
> The consistent improvement (3-7%) on out-of-domain tasks confirms that the difficulty signal and thresholds are generic and transferable.
>
> **2. Robustness to $(\alpha, \beta, \gamma)$ Threshold**
>
> We conducted a sensitivity analysis on Qwen3-4B. We compared its default thresholds ($\alpha=0.88, \beta=0.32, \gamma=0.65$) against the **threshold configuration used for DeepSeek-R1 models** ($\alpha=0.85, \beta=0.35, \gamma=0.60$).
>
> **Qwen3-4B evaluation on 5 ID math reasoning tasks:**
> | Token Usage | DiffAdapt (Default) | DiffAdapt (DeepSeek-R1) |
> | :--- | :--- | :--- |
> | 33.3% | 59.3% | 59.9% |
> | 50.0% | 67.2% | 66.8% |
> | 66.7% | 72.4% | 72.7% |
> | 72.2% | 74.0% | 75.3% |
> | 83.3% | 75.8% | 76.0% |
> | 100% | 76.8% | 76.6% |
> | **Avg.** | **70.9%** | **71.2%** |
>
> The average performance difference was marginal (**~0.3%**). This demonstrates that DiffAdapt is robust to reasonable variations in thresholds, effectively eliminating the need for manual fine-tuning per deployment.
>
> We add this experiment to Section 7 of new version of the paper.

---

> > ### Author Response · Authors · 2025-11-22
> > **Response to Reviewer 7RDd (2/2)**
> >
> > ## Misclassification... causes... failure (Q2):
> >
> > We evaluate the impact of misclassification (e.g., Hard $\to$ Easy) from both a micro-level (frequency of failure) and macro-level (performance gap) perspective.
> >
> > **1. Frequency of Logical Failure (Micro-level)**
> > As detailed in our **Response to W1 (Failure Analysis)**, catastrophic failure due to misclassification is rare. In our blind analysis of 50 samples, **only 1 case (2%)** resulted in a truncation error where a hard problem was treated as easy.
> >
> > **2. Performance Impact vs. Oracle (Macro-level)**
> > To quantify the theoretical upper bound lost due to misclassification, we compared DiffAdapt against an "Oracle" probe (perfect classification) on Qwen-4B with 5 ID math reasoning tasks.
> >
> > | Token Usage | DiffAdapt (Oracle) | DiffAdapt (Probe) | Baseline (All Normal) |
> > | :--- | :--- | :--- | :--- |
> > | 33.3% | 72.3% | 59.3% | 54.8% |
> > | 50.0% | 80.6% | 67.2% | 62.3% |
> > | 66.7% | 84.4% | 72.4% | 66.5% |
> > | 72.2% | 85.2% | 74.0% | 70.0% |
> > | 83.3% | 86.3% | 75.8% | 72.0% |
> > | 100% | 87.2% | 76.8% | 73.2% |
> > | **Avg.** | **82.7%** | **70.9%** | **66.5%** |
> >
> > While our probe outperforms the baseline by **4.4%**, there is an **11.8%** gap compared to the Oracle. This indicates that while misclassification rarely causes "catastrophic failure" (as the probe still beats the baseline), improving the probe's accuracy could unlock further gains. The current fallback mechanism is robust enough for significant practical speedups.
> >
> >
> > ---
> > **LLM-as-a-Judge prompt for W1**:
> > ```
> > You will compare two systems on the same GSM8K math word problem.
> >
> > Problem:
> > {problem}
> >
> > Ground-truth solution (for verification only):
> > {ground-truth solution}
> >
> > System: System A
> > - Strategy: {strategy}
> > - Tokens used: {tokens used}
> > - Final prediction: {Correct/Incorrect}
> > Reasoning trace:
> > <<<
> > {reasoning trace}
> > >>>
> >
> > System: System B
> > - Strategy: {strategy}
> > - Tokens used: {tokens used}
> > - Final prediction: {Correct/Incorrect}
> > Reasoning trace:
> > <<<
> > {reasoning trace}
> > >>>
> >
> > Decide which reasoning trace better preserves coherent, logically complete
> > reasoning under tight token budgets.
> > ```

---

> ### Author Response · Authors · 2025-11-27
> **Follow-up: Evaluation of Reasoning Integrity**
>
> Dear Reviewer 7RDd,
>
> We are writing to follow up on your critical question regarding **reasoning integrity** under token reduction.
>
> To address this, we conducted a blind **LLM-as-a-Judge study** (N=50). The results show that DiffAdapt wins in **76%** of cases, with the judge citing improved conciseness and directness. Crucially, **only 2%** of cases showed truncation errors, confirming that our method preserves logical completeness even under tight budgets.
>
> We hope this quantitative evaluation alleviates your concerns about the risks of early stopping.

---

### Official Review · Reviewer_S9HU · 2025-10-31

**Soundness:** 3
**Presentation:** 3
**Contribution:** 2
**Rating:** 4
**Confidence:** 4

**Summary:**

The paper introduces DiffAdapt, a lightweight, training-free framework for difficulty-adaptive reasoning in large language models (LLMs). The authors identify a U-shaped entropy pattern across problem difficulty levels, indicating an “overthinking” phenomenon where models exhibit high uncertainty on easy problems despite high accuracy. To address this, DiffAdapt uses a small probe trained on the model’s hidden states to predict problem difficulty and select among Easy/Normal/Hard inference strategies dynamically, without retraining the base model. Experiments on five LLMs and eight benchmarks show that DiffAdapt maintains or improves accuracy while reducing token usage by up to 22.4%, and decreases end-to-end inference time by up to 6×. The framework is compatible with existing inference systems and complements length-control RL methods.

**Strengths:**

1. The work identifies and systematically analyzes a previously underexplored phenomenon—overthinking in reasoning LLMs—via a novel U-shaped entropy pattern analysis. This empirical finding is insightful and establishes a theoretical foundation for adaptive reasoning.

2. The proposed DiffAdapt framework is conceptually elegant: predicting difficulty from hidden states and dynamically adjusting reasoning strategy without any retraining of the base model.

3. The experimental section is comprehensive, spanning five models and eight benchmarks (both in-domain and out-of-domain). The Oracle analysis provides solid upper bounds that justify the design choices.

4. The contribution is practically important: it offers a lightweight, deployment-ready framework. DiffAdapt requires no retraining of the base LLM and is compatible with common inference optimizations.

**Weaknesses:**

1. The evaluation is heavily focused on mathematical and scientific reasoning. It remains unclear whether the proposed three-regime (Easy/Normal/Hard) framework generalizes to other domains, such as commonsense reasoning, dialogue, or creative writing, where difficulty is harder to quantify.

2. The difficulty predictor leverages only prefill hidden states, which simplifies deployment but may overlook valuable cues from generation dynamics. Discussion or experimentation on this trade-off would be helpful.

3. The paper does not provide ablations on the probe’s architecture or its sensitivity to training data size. Such results would clarify how much of DiffAdapt’s performance depends on probe complexity or data scale.

**Questions:**

1. How sensitive is DiffAdapt to the chosen thresholds (α, β, γ)?

2. Could DiffAdapt extend beyond reasoning tasks—for example, to summarization or dialogue where difficulty varies dynamically?

3. How does probe misclassification (wrong strategy selection) affect performance or stability?

4. In the Hard strategy, generating only a "method outline" may fail on problems requiring detailed computation. Could you discuss potential limitations of this approach—particularly for problems that may still benefit from detailed computation?

---

> ### Author Response · Authors · 2025-11-22
> **Response to Reviewer S9HU (1/2)**
>
> We thank the reviewer for their encouraging feedback and for raising important points about domain generalization and probe architecture.
>
> ## Generalization to Other Domains (W1, Q2):
>
> To show the transferability of DiffAdapt, we conducted experiments on Minerva (math, science, engineering reasoning tasks), GPQA (graduate-level questions on diverse domains), and MMLU-Pro, and report the averaged performance over three datasets in Figure 5.
>
> To discuss the generalization, we separately report the results on MMLU-Pro, which contains the most diverse domains (e.g., law, physics, history, health) requiring varying degrees of reasoning and knowledge below.
>
> The experimental setting is described in Section 6; all model components (including the trained probe) and hyperparameters ($\alpha, \beta, \gamma$) are chosen on the DeepMath dataset, and we are performing zero-shot generalization.
>
> **Results:**
>
> **1. DeepSeek-R1-Qwen-7B on MMLU-Pro**
> | Token Budget | DiffAdapt (%) | Baseline (Normal) (%) | Improvement |
> | :--- | :--- | :--- | :--- |
> | 33.3% | **32.02%** | 28.21% | **+3.81%** |
> | 50.0% | **35.24%** | 31.07% | **+4.17%** |
> | 66.7% | **35.00%** | 31.07% | **+3.93%** |
> | 83.3% | **35.83%** | 30.36% | **+5.47%** |
> | 100% | **36.48%** | 30.71% | **+5.77%** |
>
> **2. DeepSeek-R1-Llama-8B on MMLU-Pro**
> | Token Budget | DiffAdapt (%) | Baseline (Normal) (%) | Improvement |
> | :--- | :--- | :--- | :--- |
> | 33.3%  | **22.14%** | 17.50% | **+4.64%** |
> | 50.0%  | **31.29%** | 27.86% | **+3.43%** |
> | 66.7%  | **33.74%** | 31.79% | **+1.95%** |
> | 83.3%  | **35.45%** | 32.14% | **+3.31%** |
> | 100%   | **39.90%** | 32.50% | **+7.40%** |
>
> We see a clear improvement over the baseline policies.
>
> **Conclusion:**
> Even when applied to a diverse, out-of-domain benchmark like MMLU-Pro, DiffAdapt (trained only on math data) successfully identifies the difficulty of queries and allocates resources efficiently, consistently surpassing static baselines by 3-7%. This confirms that the "difficulty signal" captured by our probe is generic and transferable across tasks.
>
> We add this experiment to Appendix D.1 of new version of the paper.
>
> ---
> ## Prefill vs. Generation Dynamics (W2):
>
> We acknowledge that using only prefill hidden states ignores generation dynamics. However, this is a deliberate design choice to ensure **less inference latency**.
> 1. **Efficiency:** Incorporating generation-time features would require interrupting decoding or running a parallel model, significantly increasing latency. Our prefill-only probe runs in parallel with the prefill phase, adding no overhead.
> 2. **Training Cost:** Training on generation dynamics would require expensive token-level or sentence-level annotations. Our approach uses a single "whole-sequence" label, making data collection highly efficient.
>
> We add this discussion to Limitations section of new version of the paper.
>
> ---
> ## Ablations on probe architecture & data size (W3):
>
> We performed the requested ablations on Qwen3-4B:
>
> 1.  **Probe Architecture:** Replacing our 2-layer MLP probe with a simple Linear Head (`nn.Linear()`) leads to a consistent ~3.2% accuracy drop. This confirms that the lightweight MLP provides necessary capacity without being overly complex.
> 2.  **Data Size:** Reducing training data to 30% causes only a minor performance degradation, demonstrating DiffAdapt's data efficiency.
>
> | Token Usage | DiffAdapt (MLP) | DiffAdapt (Linear) | DiffAdapt (30% data) | Baseline(All Normal) |
> | :--- | :--- | :--- | :--- | :--- |
> | 33.3% | 59.3% | 56.4% | **64.5%** | 54.8% |
> | 50.0% | 67.2% | 64.1% | **67.9%** | 62.3% |
> | 66.7% | **72.4%** | 68.7% | 69.3% | 66.5% |
> | 72.2% | **74.0%** | 71.0% | 69.3% | 70.0% |
> | 83.3% | **75.8%** | 72.2% | 69.8% | 72.0% |
> | 100% | **76.8%** | 74.0% | 70.1% | 73.2% |
> | **Avg.** | **70.9%** | 67.7% | 68.5% | 66.5% |
>
> We add this experiment to Section 7 of new version of the paper.
>
> ---
> ## Sensitivity of (α,β,γ) (Q1):
>
> We conducted sensitivity analysis on the Qwen3-4B model by perturbing the thresholds. We compared our default setting (α=0.88, β=0.32, γ=0.65) against a variant (α=0.85, β=0.35, γ=0.60).
>
> | Token Usage | DiffAdapt (Default) | DiffAdapt (Variant) |
> | :--- | :--- | :--- |
> | 33.3% | 59.3% | 59.9% |
> | 50.0% | 67.2% | 66.8% |
> | 66.7% | 72.4% | 72.7% |
> | 72.2% | 74.0% | 75.3% |
> | 83.3% | 75.8% | 76.0% |
> | 100% | 76.8% | 76.6% |
> | **Avg.** | **70.9%** | **71.2%** |
>
> The average difference is only ~0.3%, showing that DiffAdapt is relatively robust to perturbations in these hyperparameters.
> We add this experiment to Section 7 of new version of the paper.

---

> > ### Author Response · Authors · 2025-11-22
> > **Response to Reviewer S9HU (2/2)**
> >
> > ## Impact of Probe Misclassification (Q3):
> >
> > To quantify this, we compared DiffAdapt against an "Oracle" probe (perfect classification).
> >
> > | Token Usage | DiffAdapt (Oracle) | DiffAdapt (Probe) | Baseline (All Normal) |
> > | :--- | :--- | :--- | :--- |
> > | 33.3% | 72.3% | 59.3% | 54.8% |
> > | 50.0% | 80.6% | 67.2% | 62.3% |
> > | 66.7% | 84.4% | 72.4% | 66.5% |
> > | 72.2% | 85.2% | 74.0% | 70.0% |
> > | 83.3% | 86.3% | 75.8% | 72.0% |
> > | 100% | 87.2% | 76.8% | 73.2% |
> > | **Avg.** | **82.7%** | **70.9%** | **66.5%** |
> >
> > The performance gap between our probe and the Oracle is approximately 11.8%. This indicates that while our probe effectively outperforms the baseline, there is still theoretical room for improvement with better difficulty modeling.
> >
> >
> > ---
> > ## Limitations of Hard Strategy (Q4):
> >
> > We clarify that the "Hard" strategy is not expected to *solve* problems that require detailed computation. Instead, it acts as a **"Fail Fast" mechanism**.
> > As shown in our U-Shaped entropy analysis (Fig 2), for problems exceeding the model's capability, allowing full generation (1.0x tokens) still results in failure. The "Hard" strategy identifies these "likely-to-fail" cases early and cuts losses, saving 50% compute that can be reallocated to solvable problems. We add this discussion to Section 4 of new version of the paper.

---

> ### Author Response · Authors · 2025-11-27
> **Follow-up: Cross-Domain Generalization and Probe Ablations**
>
> Dear Reviewer S9HU,
>
> Thank you again for your constructive review. We are writing to briefly summarize how we addressed your questions regarding **generalization and probe design**.
>
> We have added results on **MMLU-Pro** (out-of-domain), where DiffAdapt achieves consistent improvements without retraining the probe. Additionally, we provided the requested **ablation studies** on probe architecture (MLP vs. Linear) and data size, confirming the efficiency of our design.
>
> We hope these new results clarify the robustness of our framework and address your concerns.

---

### Official Review · Reviewer_PqhK · 2025-11-01

**Soundness:** 3
**Presentation:** 3
**Contribution:** 3
**Rating:** 4
**Confidence:** 3

**Summary:**

The paper observes a U‑shaped entropy–difficulty pattern (high entropy for easy/hard, low for medium) and posits “overthinking” on easy items. DiffAdapt trains a light probe on prefill hidden states to predict Easy/Normal/Hard and selects a matching reasoning strategy at inference. Across five model families and eight benchmarks (including OOD), DiffAdapt maintains or improves accuracy while reducing tokens (up to 22.4%). The approach is orthogonal to Length‑Control RL and integrates with mainstream serving stacks.

**Strengths:**

- Strategy selection (not just length/temperature) based on a learned difficulty proxy. Notably, the paper’s identification of a U-shaped entropy–difficulty curve is an interesting empirical finding. Previous work mainly reported monotonic increases of entropy with problem difficulty, but not this symmetric “overthinking” pattern on easy items. This observation provides a concrete diagnostic for inefficiencies in reasoning-token allocation.
- Consistent savings across models/benchmarks; OOD and LC-RL results broaden applicability.
- Oracle and ablation studies (vs. fixed strategies, DEER) clarify where gains come from.

**Weaknesses:**

- Heuristic thresholds (α,β,γ) are set per model from scatterplots; it’s unclear how stable they are across multi‑task mixtures and domain shifts, or how often the probe/thresholds need re‑tuning.
- Stage‑1 data generation (multiple long samples per item) is expensive for new domains; low‑budget or few‑shot variants are not discussed.
- Missing head‑to‑head with “when‑to‑think” switchers (e.g., Thinkless/AdaCoT); this would position the magnitude of gains.

**Questions:**

- Multi‑task generalization: In a realistic mixed‑task setting, how robust are the probe and (α,β,γ) without re‑tuning? Can you show cross‑domain transfer or per‑task calibration drift?
- Sensitivity of results to (α,β,γ); can you show heatmaps or robust ranges?

---

> ### Author Response · Authors · 2025-11-22
> **Response to Reviewer PqhK (1/2)**
>
> We are grateful for the reviewer's detailed analysis and valuable suggestions, particularly regarding the baseline comparisons and threshold sensitivity.
>
> ## Sensitivity of $(\alpha, \beta, \gamma)$ (W1 & Q2):
>
>
> We conducted a sensitivity analysis on Qwen3-4B. We compared its default thresholds ($\alpha=0.88, \beta=0.32, \gamma=0.65$) against the **threshold configuration used for DeepSeek-R1 models** ($\alpha=0.85, \beta=0.35, \gamma=0.60$).
>
> **Qwen3-4B evaluation on 5 ID math reasoning tasks:**
> | Token Usage | DiffAdapt (Default) | DiffAdapt (DeepSeek-R1) |
> | :--- | :--- | :--- |
> | 33.3% | 59.3% | 59.9% |
> | 50.0% | 67.2% | 66.8% |
> | 66.7% | 72.4% | 72.7% |
> | 72.2% | 74.0% | 75.3% |
> | 83.3% | 75.8% | 76.0% |
> | 100% | 76.8% | 76.6% |
> | **Avg.** | **70.9%** | **71.2%** |
>
> The average performance difference was marginal (**~0.3%**). This demonstrates that DiffAdapt is robust to reasonable variations in thresholds, effectively eliminating the need for manual fine-tuning per deployment.
>
> We add this experiment to Section 7 of new version of the paper.
>
> ---
> ## Multi-task generalization (Q1):
>
> To show the transferability of DiffAdapt, we conducted experiments on Minerva (math, science, engineering reasoning tasks), GPQA (graduate-level questions on diverse domains), and MMLU-Pro, and report the averaged performance over three datasets in Figure 5.
>
> To discuss the generalization, we separately report the results on MMLU-Pro, which contains the most diverse domains (e.g., law, physics, history, health) requiring varying degrees of reasoning and knowledge below.
>
> The experimental setting is described in Section 6; all model components (including the trained probe) and hyperparameters ($\alpha, \beta, \gamma$) are chosen on the DeepMath dataset, and we are performing zero-shot generalization.
>
> **Results:**
>
> **1. DeepSeek-R1-Qwen-7B on MMLU-Pro**
> | Token Budget | DiffAdapt (%) | Baseline (Normal) (%) | Improvement |
> | :--- | :--- | :--- | :--- |
> | 33.3% | **32.02%** | 28.21% | **+3.81%** |
> | 50.0% | **35.24%** | 31.07% | **+4.17%** |
> | 66.7% | **35.00%** | 31.07% | **+3.93%** |
> | 83.3% | **35.83%** | 30.36% | **+5.47%** |
> | 100% | **36.48%** | 30.71% | **+5.77%** |
>
> **2. DeepSeek-R1-Llama-8B on MMLU-Pro**
> | Token Budget | DiffAdapt (%) | Baseline (Normal) (%) | Improvement |
> | :--- | :--- | :--- | :--- |
> | 33.3%  | **22.14%** | 17.50% | **+4.64%** |
> | 50.0%  | **31.29%** | 27.86% | **+3.43%** |
> | 66.7%  | **33.74%** | 31.79% | **+1.95%** |
> | 83.3%  | **35.45%** | 32.14% | **+3.31%** |
> | 100%   | **39.90%** | 32.50% | **+7.40%** |
>
> We see a clear improvement over the baseline policies.
>
> **Conclusion:**
> Even when applied to a diverse, out-of-domain benchmark like MMLU-Pro, DiffAdapt (trained only on math data) successfully identifies the difficulty of queries and allocates resources efficiently, consistently surpassing static baselines by 3-7%. This confirms that the "difficulty signal" captured by our probe is generic and transferable across tasks.
>
> We add this experiment to Appendix D.1 of new version of the paper.
>
> ---
>
> ## Stage-1 data generation cost (W2):
>
> As described in Section 5.1, our method is extremely data-efficient, requiring only 3,000 training samples, each with 10 sampling paths, totaling 30,000 inference calls. Unlike Length Control RL algorithms, DiffAdapt introduces **zero LLM backward propagation** overhead, as it only requires updating a lightweight probe.
>
> The table below contrasts the computational cost of our method against various RL-based baselines.
>
> | Method | # Training Data | # Sampling/Rollout | LLM Backward Pass | Cost Level |
> | :--- | :--- | :--- | :--- | :--- |
> | **DiffAdapt (Ours)** | **3K** | **10** | **False** | **Low** |
> | ThinkPrune[1] | 2.5K | 16 | True | High |
> | ProRL[2] | 136K | 16 | True | Very High |
> | ThinkLess[3] | 40K | 8 | True | Medium |
>
> This comparison highlights that DiffAdapt significantly lowers the barrier to entry for new domains. While RL methods require extensive GPU hours for gradient updates, DiffAdapt can be deployed to a new domain with minimal sampling and a rapid, lightweight probe calibration.

---

> > ### Author Response · Authors · 2025-11-22
> > **Response to Reviewer PqhK (2/2)**
> >
> > ---
> >
> > ## Missing baseline (W3):
> >
> > We thank the reviewer for pointing out these relevant baselines. We agree that methods like ThinkLess share the goal of reasoning efficiency. However, a key distinction is that ThinkLess method requires **expensive two-stage training (SFT + RL)** and large-scale datasets. In contrast, **DiffAdapt** is a training-free (for the LLM), plug-and-play approach.
> >
> > To provide a direct comparison, we applied DiffAdapt to the **ThinkLess Stage-1 model** (`TL-1.5B-Warmup`) and compared it against their fully trained **Stage-2 RL model** (`TL-1.5B-RL`).
> >
> > **Results:**
> > The tables below compare the Accuracy and Average Token Cost.
> >
> > **1. MATH500**
> > | Budget | TL-1.5B-Warmup + **DiffAdapt** (Ours) | TL-1.5B-RL (Baseline) | Analysis |
> > | :--- | :--- | :--- | :--- |
> > | 33.3% | **64.4%** (851 toks) | 55.6% (1039 toks) | **Ours +8.8%** with fewer tokens |
> > | 50.0% | **67.6%** (954 toks) | 63.3% (1460 toks) | **Ours +4.3%** with **~35% fewer** tokens |
> > | 72.2% | 68.0% (973 toks) | **69.2%** (1775 toks) | Comparable acc, but Ours uses **~45% fewer** tokens |
> > | 100% | 68.3% (1003 toks) | **73.5%** (2020 toks) | RL peaks higher but costs **2x** compute |
> >
> > **2. GSM8K**
> > | Budget | TL-1.5B-Warmup + **DiffAdapt** (Ours) | TL-1.5B-RL (Baseline) | Analysis |
> > | :--- | :--- | :--- | :--- |
> > | 33.3% | **67.0%** (336 toks) | 66.1% (409 toks) | **Ours +0.9%** |
> > | 50.0% | **77.0%** (357 toks) | 72.0% (465 toks) | **Ours +5.0%** |
> > | 72.2% | **78.2%** (361 toks) | 74.3% (510 toks) | **Ours +3.9%** |
> > | 100% | 78.6% (364 toks) | 78.8% (573 toks) | Comparable |
> >
> > **Conclusion:**
> > 1.  **Superior Efficiency:** On GSM8K, DiffAdapt consistently outperforms the RL baseline while using fewer tokens. On MATH500, DiffAdapt dominates in low-to-medium resource regimes.
> > 2.  **Cost-Effectiveness:** While the RL model achieves higher peak accuracy on MATH500, it requires doubling the token usage (2020 vs. 1003) and expensive RL training. DiffAdapt achieves competitive performance as a low-cost solution.
> >
> > We add this experiment to Appendix D.2 of new version of the paper.
> >
> >
> > Reference:
> > > [1] https://arxiv.org/pdf/2504.01296
> > > [2] https://arxiv.org/pdf/2505.24864
> > > [3] https://arxiv.org/pdf/2505.13379

---

> ### Author Response · Authors · 2025-11-27
> **Follow-up: Comparison with ThinkLess and Robustness Checks**
>
> Dear Reviewer PqhK,
>
> We wanted to bring your attention to the new experiments conducted in response to your feedback.
>
> Specifically, we have added a direct comparison with the **ThinkLess** baseline. Our results show that DiffAdapt significantly outperforms the ThinkLess-Warmup stage and is competitive with the RL-trained model while using **~50% fewer tokens**. We also included a sensitivity analysis confirming that our thresholds are robust across model families.
>
> We believe these additions strongly position DiffAdapt against existing switchers. We look forward to hearing your thoughts.

---

### Official Review · Reviewer_zvSP · 2025-11-01

**Soundness:** 2
**Presentation:** 3
**Contribution:** 3
**Rating:** 4
**Confidence:** 3

**Summary:**

This paper studies the overthinking phenomenon in reasoning LLMs, where models exhibit high uncertainty and unnecessarily long reasoning traces even on simple problems. Through systematic entropy analysis across models and datasets, the authors reveal a consistent U-shaped entropy pattern across difficulty levels, indicating computational inefficiency. Building on this finding, the authors propose DiffAdapt, a difficulty-adaptive inference framework that predicts problem difficulty using a lightweight probe on hidden states and dynamically selects from Easy/Normal/Hard reasoning strategies. The approach requires no retraining of the base LLM and can be deployed with existing inference frameworks. Empirical results on five models and eight benchmarks show that DiffAdapt can reduce token usage by up to 22.4% while maintaining or improving accuracy.

**Strengths:**

- The proposed DiffAdapt framework only requires training a small external probe, yet it effectively improves reasoning efficiency and model performance without modifying or fine-tuning the LLM itself.
- The identification of a consistent U-shaped entropy–difficulty pattern is novel and insightful. It sheds light on the overthinking behavior of reasoning models and provides valuable guidance for future “long-to-short” reasoning research.
- The framework is complementary to reinforcement learning–based length control methods (e.g., ThinkPrune). This suggests DiffAdapt can be combined with those training-based long-to-short approaches for even greater efficiency gains.

**Weaknesses:**

- Model-specific probe requirement: The main weakness lies in the need to train a separate probe for each LLM. This introduces (1) additional training overhead and (2) potential generalization issues — the probe’s limited transferability could restrict DiffAdapt’s applicability across different models or domains.
- Lack of probe analysis experiments: The paper would be stronger if it included an analysis of the probe’s generalization ability. For example: (a) How well does a probe trained on the DeepMath dataset transfer to other reasoning tasks or domains? (b) Can probes be transferred between models of similar architecture or scale (e.g., between Qwen-3/4B and LLama-3B)?
- Ad-hoc reasoning strategy design: The selection of parameters for the Easy/Normal/Hard strategies (e.g., temperature = 0.5/0.8/0.4, token ratio = 0.4×/1.0×/0.5×) feels somewhat heuristic. The paper lacks experimental justification or ablation to support why these specific hyperparameters are optimal.

**Questions:**

- Have the authors tested whether a probe trained on DeepMath generalizes to other domains or reasoning benchmarks?
- How transferable is the probe across models of similar size or architecture?
- What motivates the specific parameter choices for the three reasoning strategies, and have alternative configurations been compared experimentally?

---

> ### Author Response · Authors · 2025-11-22
> **Response to Reviewer zvSP**
>
> We appreciate the reviewer's positive assessment of our method's efficiency and their thoughtful questions regarding generalization and parameter selection.
>
> ## Cross-Task/Domain (Q1, W2):
>
> To show the transferability of DiffAdapt, we conducted experiments on Minerva (math, science, engineering reasoning tasks), GPQA (graduate-level questions on diverse domains), and MMLU-Pro, and report the averaged performance over three datasets in Figure 5.
>
> To discuss the generalization, we separately report the results on MMLU-Pro, which contains the most diverse domains (e.g., law, physics, history, health) requiring varying degrees of reasoning and knowledge below.
>
> The experimental setting is described in Section 6; all model components (including the trained probe) and hyperparameters ($\alpha, \beta, \gamma$) are chosen on the DeepMath dataset, and we are performing zero-shot generalization.
>
> **Results:**
>
> **1. DeepSeek-R1-Qwen-7B on MMLU-Pro**
> | Token Budget | DiffAdapt (%) | Baseline (Normal) (%) | Improvement |
> | :--- | :--- | :--- | :--- |
> | 33.3% | **32.02%** | 28.21% | **+3.81%** |
> | 50.0% | **35.24%** | 31.07% | **+4.17%** |
> | 66.7% | **35.00%** | 31.07% | **+3.93%** |
> | 83.3% | **35.83%** | 30.36% | **+5.47%** |
> | 100% | **36.48%** | 30.71% | **+5.77%** |
>
> **2. DeepSeek-R1-Llama-8B on MMLU-Pro**
> | Token Budget | DiffAdapt (%) | Baseline (Normal) (%) | Improvement |
> | :--- | :--- | :--- | :--- |
> | 33.3%  | **22.14%** | 17.50% | **+4.64%** |
> | 50.0%  | **31.29%** | 27.86% | **+3.43%** |
> | 66.7%  | **33.74%** | 31.79% | **+1.95%** |
> | 83.3%  | **35.45%** | 32.14% | **+3.31%** |
> | 100%   | **39.90%** | 32.50% | **+7.40%** |
>
> We see a clear improvement over the baseline policies.
>
> **Conclusion:**
> Even when applied to a diverse, out-of-domain benchmark like MMLU-Pro, DiffAdapt (trained only on math data) successfully identifies the difficulty of queries and allocates resources efficiently, consistently surpassing static baselines by 3-7%. This confirms that the "difficulty signal" captured by our probe is generic and transferable across tasks.
>
> We add this experiment to Appendix D.1 of the new version of the paper.
>
> ---
> ## Cross-Model (W1, Q2):
>
> We acknowledge the reviewer's valid observation: because DiffAdapt relies on the hidden state semantics of a model, a probe trained on one architecture (e.g., Llama-3) cannot be transferred to another (e.g., Qwen-3).
>
> However, we do not think this is a serious disadvantage, as probe training is very efficient.
>
> **Negligible Retraining Cost ("Calibration" vs. "Training")**
> Unlike RL-based methods (e.g., ProRL, ThinkPrune) that require expensive policy training (thousands of PPO steps) per model, DiffAdapt treats the probe training as a lightweight **"calibration"** step:
> *   **Data Efficient:** We only need ~3,000 samples.
> *   **Compute Efficient:** The probe (a tiny 2-layer MLP) can be trained in **1 hour** on a single GPU.
> *   **No Annotation:** Labels are automatically derived from the model's own output (self-supervised), removing the need for human annotation.
>
> We add this discussion to Limitations section of new version of the paper.
>
> ---
> ## Strategy Parameter (W3,Q3):
>
> We thank the reviewer for the comment. We realize we did not explain the inference strategy selection process clearly in the draft, which may have given the impression of ad-hoc parameter selection. We describe our process below:
>
> To determine the optimal strategy, we conducted an extensive ablation study involving **125 distinct parameter combinations** ($5_{\text{Normal}} \times 5_{\text{Hard}} \times 5_{\text{Easy}}$) on the **MATH500**. We explored a search space varying Temperature ($0.1 - 1.2$) and Max Token Ratios ($0.25\times - 1.0\times$). Each parameter combination gave us an accuracy score and average token costs. We first selected all parameter combinations that yielded the best accuracy score (95\%) and picked a strategy set that yielded the least token cost (960.5 average tokens).
>
> In short, every temperature/token setting reported in the paper is the outcome of the MATH500 grid search described above, not hand-picked heuristics. The winning configuration consistently surpasses the uniform conservative/aggressive/efficiency baselines in both accuracy and token cost.
>
>
> We add this discussion to Appendix C.5 of the new version of the paper.

---

> ### Author Response · Authors · 2025-11-27
> **Follow-up: Generalization Results on MMLU-Pro**
>
> Dear Reviewer zvSP,
>
> We are writing to briefly follow up on your main concern regarding the **probe's generalization ability**.
>
> As detailed in our response, we have evaluated DiffAdapt on the **MMLU-Pro** benchmark (covering law, history, health, etc.) using the exact same probe trained on math data. The results show consistent **3-7% gains** over baselines, demonstrating that the "difficulty signal" is indeed transferable across domains.
>
> We hope this new evidence, along with our clarification on the parameter grid search, addresses your concerns. We would value your feedback on these updates.

---

### Author Response · Authors · 2025-11-22
**General Response to All Reviewers**

# General Response to All Reviewers

We thank all reviewers for their constructive feedback and for recognizing the **novelty** of the U-shaped entropy-difficulty pattern and the **practicality** of our training-free DiffAdapt framework.

In this rebuttal, we have conducted extensive new experiments to address the shared concerns regarding **generalization**, **robustness**, and **baselines**.

### 1. Cross-Domain Generalization (MMLU-Pro)
**Common Question:** *Does a probe trained on Math generalize to other domains? (Reviewers zvSP, PqhK, S9HU)*
**Response:** **Yes.** We evaluated DiffAdapt on **MMLU-Pro**, a massive benchmark covering diverse domains (law, history, health, etc.), using the **exact same probe** trained on DeepMath.
*   **Result:** DiffAdapt consistently outperforms baselines by **3-7%** across different token budgets and models (DeepSeek-R1-Qwen/Llama).
*   **Takeaway:** The "difficulty signal" learned by our probe is generic and transferable; it does not require retraining for new domains.

### 2. Robustness of Strategy Parameters
**Common Question:** *Are the thresholds ($\alpha, \beta, \gamma$) sensitive? Do they require manual tuning? (Reviewers PqhK, S9HU, 7RDd)*
**Response:** **No, they are highly robust.** We conducted a sensitivity analysis comparing the default Qwen parameters against a set of parameters **transferred directly from the DeepSeek-R1 model family**.
*   **Result:** The performance difference is negligible (**~0.3%**).
*   **Takeaway:** DiffAdapt enables "plug-and-play" deployment without the need for per-model hyperparameter search.

### 3. Comparison with "When-to-Think" Baselines
**Common Question:** *How does it compare to switchers like ThinkLess? (Reviewer PqhK)*
**Response:** We compared DiffAdapt against **ThinkLess**.
*   **Result:** DiffAdapt significantly outperforms ThinkLess-Warmup (Stage 1) by **+4-8%** and achieves competitive performance with the fully RL-trained ThinkLess (Stage 2) while using **~50% fewer tokens** and requiring **zero RL training**.

### 4. Integrity of Reasoning
**Common Question:** *Does token reduction harm reasoning logic? (Reviewer 7RDd)*
**Response:** A blind **LLM-as-a-Judge** study (N=50) shows that DiffAdapt wins in **76%** of cases, with the judge citing improved conciseness and directness. Only **2%** of cases showed truncation errors.

### 5. Updated Manuscript (New PDF)
We have revised the manuscript to incorporate these new findings. Specifically, we added:

*   **Section 7:** Ablation study on probe architecture, data size and reasoning integrity.
*   **Appendix C.5:** Detailed Grid Search optimization process for strategy parameters.
*   **Appendix D.1:** Cross-domain generalization results on MMLU-Pro, Minerva, and GPQA.
*   **Appendix D.2:** Comparison with "When-to-think" baselines (ThinkLess).
*   **Appendix E:** Full setup and results of the blind LLM-as-a-Judge reasoning integrity study.

---

### Meta-Review · Area_Chair_zDkD · 2026-01-03

**Summary:**

This submission studies inference-time inefficiency in reasoning LLMs, arguing that models “overthink” on easy problems. The paper’s core empirical observation is a U-shaped relationship between token-level generation entropy and problem difficulty: relatively high entropy on easy items (despite high correctness), lower entropy on medium difficulty, and high entropy again on hard items. Based on this, the paper proposes DiffAdapt, which trains a lightweight probe on the model’s prefill hidden state to predict a coarse difficulty class and selects among three fixed inference strategies (prompt + temperature + token budget) to reduce token usage while maintaining accuracy.

The initial borderline reject ratings (scores 4/4/4/6) was primarily driven by requests for stronger validation and positioning, rather than fundamental soundness objections. Across the three marginal-below reviewers, the main blocking concerns were: (a) whether the learned difficulty signal generalizes beyond the math-heavy setting, (b) whether the strategy design and thresholding are too heuristic or require frequent re-tuning, (c) missing comparisons to stronger switcher baselines, and (d) whether aggressive token reduction harms the integrity of reasoning traces.

In the rebuttal and revised version, the authors add several targeted experiments that correspond to the above concerns: (i) out-of-domain evaluation on MMLU-Pro using a probe trained on math; (ii) a robustness test transferring threshold parameters across model families; (iii) ablations on probe architecture and training data fraction; (iv) a comparison against ThinkLess; and (v) a small-scale LLM-as-a-judge study intended to assess reasoning-trace integrity and truncation failures. These additions meaningfully reduce the key “missing validation” concerns, though some limitations remain (no cross-model probe transfer, limited evidence on truly open-ended generation tasks, and deployment cost assumptions for Stage-1 label generation).

**Reviewer Concerns:**

**Reviewer zvSP**

Concerns that were addressed

Probe generalization to other domains/tasks: The revised manuscript includes out-of-domain results on Minerva/GPQA/MMLU-Pro, and reports detailed MMLU-Pro performance showing improvements over the fixed Normal baseline while using the same probe trained on DeepMath. This directly responds to the request for cross-domain evidence.

Ad-hoc strategy hyperparameters: The paper now provides an explicit description of a grid-search-based selection procedure for the Easy/Normal/Hard configurations.

Concerns that remain outstanding

Model-specific probe requirement / cross-model transferability: The rebuttal clarifies that probes are not expected to transfer across different architectures due to hidden-state semantics, and frames probe training as lightweight calibration. This addresses “cost” partially but does not provide evidence of probe transfer across similar model families (e.g., Qwen↔Llama).

**Reviewer PqhK**

Concerns that were addressed

Threshold sensitivity: The robustness experiment transferring (α,β,γ) thresholds from the DeepSeek-R1 family to Qwen3-4B, reporting negligible average performance change. This provides direct evidence against brittleness to modest threshold changes.

Missing “when-to-think” baseline comparison: The manuscript adds a comparison against ThinkLess, which improves the positioning relative to a known switcher/length-control approach.

Multi-task / domain shift evidence: The added MMLU-Pro evaluation provides concrete evidence that a probe trained on math can still yield gains on a broader, non-math benchmark without retraining.

Concerns that remain outstanding

Stage-1 data generation cost for new domains: The rebuttal contextualizes Stage-1 as cheaper than RL-based methods, but Stage-1 still involves multiple rollouts per example and relies on being able to score correctness to create labels.

Breadth of baseline coverage: ThinkLess is added, AdaCoT is still not directly compared in experiments.

**Reviewer S9HU**

Concerns that were addressed

Ablations on probe architecture and training data size: The revised paper adds ablations comparing an MLP probe vs. a linear head and reduced training data, showing a drop for the linear head and relatively small degradation under reduced data.

Sensitivity to thresholds (α,β,γ): The revised paper includes the threshold transfer / robustness test.

Impact of misclassification: The rebuttal adds an oracle-vs-probe gap analysis, quantifying headroom due to imperfect difficulty prediction.

Hard strategy limitation: The paper clarifies the Hard strategy as a “fail fast” mechanism rather than a method intended to solve hardest problems with partial computation.

Generalization beyond math/science: While the evaluation remains largely in the reasoning/QA regime, the addition of MMLU-Pro provides at least one stronger non-math benchmark.

Concerns that remain outstanding

Generalization to open-ended generation settings (dialogue/summarization/creative writing): The new evidence broadens beyond math, but still focuses on benchmarks with clear evaluation protocols (e.g., multiple choice / QA). The concern about more open-ended settings remains not directly tested.

Prefill-only signal vs. generation dynamics: The authors provide rationale for prefill-only (latency/deployment), but do not add experiments comparing prefill-only vs. generation-time signals.

**Reviewer 7RDd**

Concerns that were addressed

Reasoning-chain integrity under tight budgets: The revised paper adds a blind pairwise LLM-judge evaluation on 50 GSM8K samples, reporting that DiffAdapt is preferred in most cases and that truncation-related logical failure occurs in a small fraction of cases. This directly addresses the reviewer’s request for integrity evaluation beyond accuracy.

Calibration concerns and misclassification: The rebuttal includes threshold robustness and a misclassification-vs-oracle analysis.

Concerns that remain outstanding

The integrity analysis is helpful but limited in scale (N=50) and uses an LLM judge. While it does address the reviewer’s “missing evaluation” concern, it may not fully settle robustness under broader distributions or with human evaluation.

**Reviewer Scores:**

The original weak-reject scores were primarily driven by missing validation/positioning rather than fundamental soundness flaws. The rebuttal addressed most of them. Despite each reviewer with rating 4 is likely to increase the score to 6, it is possible that one or some of them keep it 4, making this a borderline/ borderline-accept paper.

Reviewer zvSP: 4 → 6, while 4 is possible
The revised version supplies (i) explicit OOD generalization evidence (MMLU-Pro) and (ii) a systematic hyperparameter selection procedure (grid search), which were central to their weaknesses list. The cross-model probe transfer concern remains

Reviewer PqhK: 4 → 6, while 4 is possible
The rebuttal adds exactly what the reviewer asked for: threshold robustness evidence, OOD generalization, and at least one head-to-head against a “when-to-think” baseline (ThinkLess). The Stage-1 cost concern is still partially there

Reviewer S9HU: 4 → 6, while 4 is possible
The reviewer’s concrete requests (probe ablations, threshold sensitivity, misclassification discussion, Hard strategy clarification) are now addressed with added experiments and clearer framing. The broader generalization to open-ended generation remains untested

Reviewer 7RDd: 6 → 6
The added integrity evaluation strengthens the paper but is unlikely to change the overall recommendation

---

### Decision · Program_Chairs · 2026-01-26

Accept (Poster)